# Differential Coding for Training-Free ANN-to-SNN Conversion

**Zihan Huang** [1]   **Wei Fang**[⊠ 2]   **Tong Bu** [1]   **Peng Xue** [3 4]   **Zecheng Hao** [1]   **Wenxuan Liu** [1]   **Yuanhong Tang**[⊠ 1]
**Zhaofei Yu** [1 5]   **Tiejun Huang** [1]

## Abstract

Spiking Neural Networks (SNNs) exhibit significant potential due to their low energy consumption. Converting Artificial Neural Networks (ANNs) to SNNs is an efficient way to achieve high-performance SNNs. However, many conversion methods are based on rate coding, which requires numerous spikes and longer time-steps compared to directly trained SNNs, leading to increased energy consumption and latency. This article introduces differential coding for ANN-to-SNN conversion, a novel coding scheme that reduces spike counts and energy consumption by transmitting changes in rate information rather than rates directly, and explores its application across various layers. Additionally, the threshold iteration method is proposed to optimize thresholds based on activation distribution when converting Rectified Linear Units (ReLUs) to spiking neurons. Experimental results on various Convolutional Neural Networks (CNNs) and Transformers demonstrate that the proposed differential coding significantly improves accuracy while reducing energy consumption, particularly when combined with the threshold iteration method, achieving state-of-the-art performance. The source codes of the proposed method are available at https://github.com/h-z-h-cell/ANN-to-SNN-DCGS.

---

[1]School of Computer Science, Peking University, Beijing, China [2]School of Electronic and Computer Engineering, Shenzhen Graduate School, Peking University, Shenzhen, China [3]Peng Cheng Laboratory, Shenzhen, China [4]Shenzhen Institute of Advanced Technology, Chinese Academy of Sciences, Shenzhen, China [5]Institute for Artificial Intelligence, Peking University, Beijing, China. Correspondence to: Wei Fang <fwei@pku.edu.cn>, Yuanhong Tang <ydtang@pku.edu.cn>.

*Proceedings of the $42^{nd}$ International Conference on Machine Learning*, Vancouver, Canada. PMLR 267, 2025. Copyright 2025 by the author(s).

## 1. Introduction

Spiking Neural Networks (SNNs) are sometimes regarded as the third generation of neural network models (Maass, 1997) for their unique neural dynamics and high biological plausibility (Gerstner et al., 2014), making them a competitive candidate to Artificial Neural Networks (ANNs) (Li et al., 2024). A significant difference between ANNs and SNNs is the information representation. ANNs transmit dense floating values between layers. While in SNNs, communications between layers are based on sparse and binary spikes, which are triggered by the membrane potentials of spiking neurons across the threshold, bringing event-driven computations and extremely low power consumption on neuromorphic chips (Merolla et al., 2014; Davies et al., 2018; DeBole et al., 2019; Pei et al., 2019).

However, the discrete and non-differentiable spike firing process causes huge learning challenges in SNNs. Recently, this issue is solved partly by the surrogate learning method (Neftci et al., 2019), which redefines the gradient of spike firing process by a smooth and differentiable surrogate function. Enabled by the surrogate gradients, deep SNNs can be trained by powerful backward propagation and gradient descent methods, and their performance are greatly improved (Fang et al., 2021; Duan et al., 2022; Shi et al., 2024). The applications of SNNs are also extended to complex event-based vision tasks (Cordone et al., 2022; Liu et al., 2024; Chen et al., 2024; Liu et al., 2025). Unfortunately, the surrogate gradient method is a coarse approximation, and may mislead the gradient descent direction in multi-layer SNNs (Gygax & Zenke, 2024). The time dimension of SNNs leads to the employment of backpropagation through time (BPTT), which requires nearly $T$ times of training resources than ANNs. Here $T$ is the sequence-length, and is also the number of time-steps of SNNs. Although some online learning methods (Xiao et al., 2022; Bohnstingl et al., 2023; Meng et al., 2023; Zhu et al., 2024) can estimate the full gradients of BPTT by accumulation of single-step gradients, their task accuracy is sub-optimal.

In addition to the surrogate gradient methods, the ANN to SNN conversion methods (Cao et al., 2015; Han et al., 2020; Li et al., 2021; Deng & Gu, 2021; Bu et al., 2022a; 2024) are another spiking deep learning methodology that

eliminates training challenges of SNNs. They convert pre-trained ANNs to SNNs with replacing nonlinear activation functions by spiking neurons. The converted SNNs enjoy high performance and close accuracy to the source ANNs, even in the complex ImageNet dataset. Most of the conversion methods are based on the rate coding, which represents activations in ANNs by the firing rates of SNNs. However, the precise estimation of firing rates requires a large number of time-steps, resulting in the obviously higher latency and energy consumption of the conversion methods than the surrogate gradient methods.

In this article, we propose the differential coding and its implementation scheme for different layers in ANN-to-SNN conversion. Instead of considering the average firing rate as the encoded activation value, differential coding treats time-weighted spikes as corrections to the encoded activation value. This approach not only improves network accuracy but also allows neurons to stop firing once a certain approximation precision is achieved, thereby reducing energy consumption without any extra training. Additionally, by minimizing the expected error between the Rectified Linear Units (ReLUs) and the encoded values in SNNs, we propose the threshold iteration method to determine the optimal thresholds for the spiking neurons for converting ReLUs, further enhancing the performance of the SNN.

Our main contributions are summarized as follows:

- We propose differential coding for ANN-to-SNN conversion and establish the dynamics for various modules in SNNs.

- We design a threshold iteration method to determine the optimal thresholds of spiking neurons for converting ReLUs.

- We designed two equivalent implementations of the employed multi-threshold (MT) neuron to facilitate hardware-friendly execution.

- By converting different CNNs and Transformers into SNNs for evaluation, our extensive experiments demonstrate that the proposed method achieves state-of-the-art accuracy while significantly reducing network energy consumption.

## 2. Related Works

### 2.1. Rated-based ANN to SNN Conversion

The rate coding method has been early found in biological neural systems (Adrian, 1926) that stronger stimulation causes more frequent spikes. This straightforward coding method builds the bridge between activations of ReLUs in ANNs and firing rates of Integrate-and-Fire (IF) neurons in SNNs, based on which the primary ANN to SNN conversion

method (Cao et al., 2015) was derived. As the firing rate is defined by the average number of spikes over all time-steps, its range is restricted between zero and one. For the negative part, the IF neurons perfectly fit ReLUs. While the outputs of ReLUs are unbound, the normalization of weights for regulating activations (Rueckauer et al., 2017) and the balancing of thresholds for spiking neurons (Han et al., 2020) are proposed and relieve the range of mismatch during conversions.

The time is discretized to time-steps in SNNs. Consequently, the firing rates are also rounded with the fixed interval. While the floating activations in source ANNs are continuous, the discrete firing rates can not fit them preciously, causing the quantization errors. To future reduce the conversion errors, some quantized ANN to SNN methods are proposed (Bu et al., 2022b; Hu et al., 2023). These methods quantize and clip activations of ANNs, relieving the quantization errors and range mismatch at the same time. However, the source ANNs must be re-trained, which increases conversion costs, and their performance declines due to the change in the activation function.

The spikes may not arrive evenly during inference, which may cause the unevenness error in conversion (Bu et al., 2022b). A typical case is that no spike during the first $\frac{T}{2}$ time-steps, and more than $\frac{T}{2}$ spikes from different synapses arrive at the neuron during the last $\frac{T}{2}$ time-steps. Although the input firing rate is larger than 0.5, the neuron can not generate the 0.5 output firing rate because it has not enough time-steps to fire. Several methods have been proposed to reduce this error, including the two-stage inference strategy (Hao et al., 2023a) and shifting the initial membrane potential (Hao et al., 2023b).

Recent research has been extended to convert ANNs with activations beyond ReLUs to SNNs. (Oh & Lee, 2024) introduced a sign gradient descent based neuron that can approximate various nonlinear activation functions. (Wang et al., 2023) and (kang you et al., 2024) trained modified Transformers and converted them into spiking Transformers. Meanwhile, (Jiang et al., 2024) and (Huang et al., 2024) developed modules to approximate nonlinear layers, enabling a training-free conversion of Transformers to SNNs.

### 2.2. Temporal Coding Conversions

The rate coding method is inefficient and causes huge latency in rate-based ANN to SNN conversion methods. The surrogate gradient methods avoid this issue by the end-to-end training, while the interpretation of coding methods in these SNNs is not clear yet (Li et al., 2023). For the conversion method, manual design for the coding strategy is indispensable. Beyond the rate coding method, several temporal coding methods are explored.

The time-to-first-spike (TTFS) coding method (Rueckauer & Liu, 2018; Zhang et al., 2019; Stanojevic et al., 2023) encodes the value into the firing time of spikes. Each neuron only fires one spike in TTFS SNNs, which brings extremely high power efficiency. However, these methods rely on layer-by-layer processing, i.e., one layer can only start to compute after it receives all input spikes from the last layer. Thus, these TTFS SNNs suffer from high latency increasing with the network depth.

The phase coding methods (Kim et al., 2018; Wang et al., 2022b) are similar to binary/decimal conversion. They encode values from spikes with the power-of-2 weights. For a given number of time-steps $T$, these methods can represent $2^T$ different values, while the rate coding method can only represent $T$ values. Their drawbacks are similar to the latency problem in TTFS SNNs that the values can only be obtained after all weighted spikes in a phase arrive.

The burst coding methods (Park et al., 2019; Li & Zeng, 2022; Wang et al., 2025) imitates the bursts of spikes during a short period in biological neural systems. The burst spikes are implemented by the multiplication of spikes and a coefficient, which carry more information than binary spikes, but may lose the advantages of SNNs based on the binary characteristic.

## 3. Preliminaries

### 3.1. Multi-Threshold Neuron

Many previous works have proposed using ternary-valued neurons to simulate negative values and reduce conversion errors (Li et al., 2022; Wang et al., 2022a; kang you et al., 2024). The ternary representation, with outputs of -1, 0, and 1, does not disrupt the event-driven nature and significantly enhances the expressive capability. Furthermore, (Huang et al., 2024) introduced the use of multi-channel methods to implement multi-threshold (MT) neuron for spike communication. In this article, we similarly adopt this approach to simulate $y = x$ using identity spiking MT neurons.

The MT neuron is characterized by several parameters, including the base threshold $\theta$, and a total of $2n$ thresholds, with $n$ positive and $n$ negative thresholds. The threshold values of the MT neuron are indexed by $i$, where $\lambda_i^l$ represents the $i$-th threshold value in the layer $l$:

$$\lambda_1^l = \theta^l, \lambda_2^l = \frac{\theta^l}{2}, ..., \lambda_n^l = \frac{\theta^l}{2^{n-1}},$$
$$\lambda_{n+1}^l = -\theta^l, \lambda_{n+2}^l = -\frac{\theta^l}{2}, ..., \lambda_{2n}^l = -\frac{\theta^l}{2^{n-1}}. \tag{1}$$

Let variables $I^l[t]$, $W^l$, $s_i^l[t]$, $x^l[t]$, $m^l[t]$, and $v^l[t]$ represent the input current, weight, the output spike of the $i$-th threshold, the total output signal, and the membrane potential before and after spikes in the $l$-th layer at the time-step

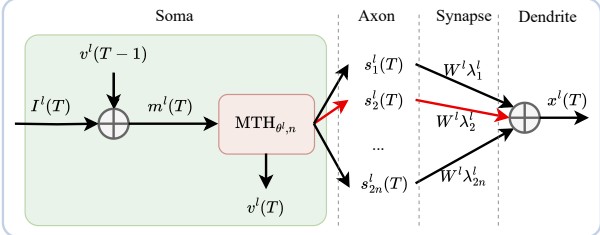

*Figure 1.* Diagram of the MT neuron. The MT neuron receives input from last module and emits up to one spike at each time-step.

$t$. The dynamics of the MT neurons are described by the following equations:

$$m^l[t] = v^l[t-1] + I^l[t] = v^l[t-1] + x^{l-1}[t], \tag{2}$$
$$s_i^l[t] = \text{MTH}_{\theta,n}(m^l[t], i) \tag{3}$$
$$x^l[t] = \sum_i s_i^l[t] W^l \lambda_i^l, \tag{4}$$
$$v^l[t] = m^l[t] - x^l[t], \tag{5}$$

$$\text{MTH}_{\theta,n}(m^l[t], i) = \begin{cases} 0, \text{if} \quad \lambda_{2n} < x < \lambda_n \\ 1, \text{elif} \quad i = \arg\min_p |x - \lambda_p| \\ 0, \text{else} \end{cases}. \tag{6}$$

Figure 1 shows the dynamics of MT neurons, when $n = 1$, this model reduces to an IF neuron with an additional negative threshold. Since only up to one threshold can emit spike per time-step, and $\lambda^l$ can be derived by bit-shifting $\theta$, we can implement MT neurons by calculating $W^l \lambda_i^l$ through the weight $W^l \theta^l$ followed by bit-shifting.

### 3.2. Rate Coding in ANN-to-SNN Conversion

Traditional ANN-to-SNN conversion methods employ rate coding, which can be mathematically expressed as:

$$r^l[t] = \frac{1}{t} \sum_{i=1}^{t} x^l[i], \tag{7}$$

where $x^l[i]$ represents the encoded output of layer $l$ in SNNs at time-step $i$, while $r^l[t]$ denotes the encoded activation that aims to map activation value $\alpha^l$ from the corresponding ANN layer. Derived from Equation (2) and (5), we have

$$v^l[t] - v^l[t-1] = x^{l-1}[t] - x^l[t], \tag{8}$$
$$v^l[t] - v^l[0] = \sum_{i=1}^{t} x^{l-1}[t] - \sum_{i=1}^{t} x^l[t], \tag{9}$$
$$r^l[t] = \frac{1}{t} \sum_{i=1}^{t} x^l[t] = \frac{1}{t} \sum_{i=1}^{t} x^{l-1}[t] - \frac{v^l[t] - v^l[0]}{t}$$
$$= r^{l-1}[t] - \frac{v^l[t] - v^l[0]}{t}. \tag{10}$$

Assuming $r^{l-1}[t] = \alpha^{l-1}$, and given that $\alpha^l = \alpha^{l-1}$ when simulating $y = x$, when $t$ is sufficient large or $v^l[t]$ close to $v^l[0]$, the encode value $r^l[t]$ in SNNs can approximate the activation value $\alpha^l$ in ANNs:

$$r^l[t] = r^{l-1}[t] - \frac{v^l[t] - v^l[0]}{t}$$
$$\approx r^{l-1}[t] = \alpha^{l-1} = \alpha^l. \tag{11}$$

## 4. Method

In this section, we propose differential coding with graded units and spiking neurons (DCGS), a training-free theory for converting ANNs to SNNs. We begin by introducing a differential coding approach, from which we develop differential graded units, differential spiking neurons and differential coding for linear Layer. These enable the conversion of various network modules. Additionally, we provide the threshold iteration method to find the optimal threshold of spiking neurons for converting ReLUs. The overall algorithm can be found in Appendix A. Furthermore, we design two equivalent implementations of the MT neuron to support hardware-friendly execution.

### 4.1. Differential Coding in ANN-to-SNN Conversion

The traditional ANN-to-SNN conversion uses rate coding to transmit information, where the firing rate $r^l[t]$ at each time-step encodes the activation value. Equation (12) shows the relationship between the output firing rate $r^l[t]$ and the output signal $x^l[t]$. When layer $l$ consists of spiking neurons, the state can be described as $x^l[t] = \theta^l s^l[t]$, where $s^l[t]$ denotes the spike at time-step $t$ and $\theta^l$ represents the threshold.

$$r^l[t] = \frac{1}{t} \sum_{i=1}^{t} x^l[t] = \frac{t-1}{t} r^l[t-1] + \frac{x^l[t]}{t}. \tag{12}$$

When the neuron does not emit a spike, the rate update is the proportion $-\frac{1}{t} r^l[t-1]$ of the previous rate, while when the neuron emits a spike, the rate update increases by $-\frac{1}{t} r^l[t-1] + \frac{x^l[t]}{t}$.

However, the rate coding method has a problem: over time, the encoded value gradually decays. As the time-step $t$ increases, the influence of earlier inputs $\frac{1}{t}$ becomes smaller, and the system requires more spikes to compensate for this decay effect, thus increasing the number of spikes required.

To address this issue, we propose a novel encoding scheme, referred to as differential coding.

**Definition 4.1.** In differential coding, denote $x^l[t]$ as the actual output of the neuron. Define $e^l[t]$ as the encoded output value at time-step $t$, the encoded activation value $r^l[t]$ as the average of $e^l[t]$ from time 1 to time-step $t$. The

relationship between the two is expressed by Equations (13) and (14), as follows:

$$e^l[t] = r^l[t-1] + x^l[t], \tag{13}$$

$$r^l[t] = r^l[t-1] + \frac{x^l[t]}{t} = \frac{1}{t} \sum_{i=1}^{t} e^l[i], \tag{14}$$

where $t$ starts from 1, $r^l[0] = 0$.

The detailed explanation of Definition 4.1 is provided in the Appendix B. Comparing Equation (7) with (14), the key difference is that differential coding only updates the encoded activation value when an output spike occurs, rather than decay at each time-step in rate coding. Figure 2 shows the ideal fitting results of rate coding and differential coding for Input $y = x$ with $T = 3$ and thresholds $\pm 1$. Differential coding can represent a wider range of values and achieve higher precision than rate coding, given the same threshold and time-steps.

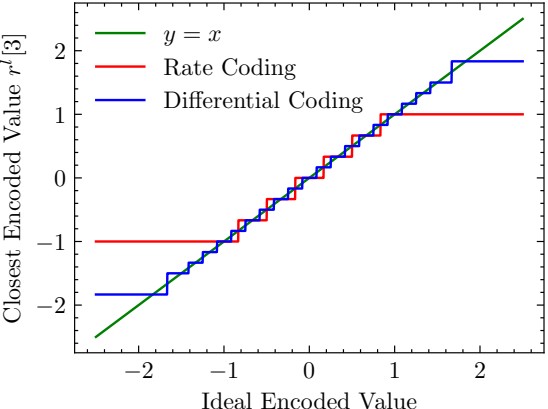

*Figure 2.* Comparison of ideal fitting results: rate coding vs. differential coding for input $y = x$ with $T = 3$ and thresholds $\pm 1$. Differential coding shows a wider representation range and higher precision.

#### 4.1.1. DIFFERENTIAL GRADED UNITS

Existing ANN-to-SNN conversion methods struggle with nonlinear functions such as Gaussian Error Linear Units (GeLU) (Hendrycks & Gimpel, 2023) and LayerNorm (Lei Ba et al., 2016). For these nonlinear layers, we utilize specific neuron dynamic units to implement them. Based on the expectation compensation idea from (Huang et al., 2024), we propose introducing differential graded units to replace those nonlinear modules that cannot be directly converted.

Derived from differential coding scheme in Definition 4.1, this article proposes two types of differential graded units. Theorem 4.2 corresponds to nonlinear layers with only one input $x^{l-1}[t]$, and Theorem 4.3 applies to certain operations $\cdot$ with two inputs $x_A^{l-1}[t]$ and $x_B^{l-1}[t]$.

**Theorem 4.2.** *Let $F^l$ be a nonlinear layer $l$ with only one input $\boldsymbol{x}^{l-1}[t]$, such as Gelu, Silu, Maxpool, LayerNorm, or Softmax. In ANN-to-SNN conversion, the mapping from $F$ to dynamics of the differential graded unit in differential coding is given by Equations (15) and (16).*

$$\boldsymbol{m}^l[t] = \boldsymbol{r}^{l-1}[t] = \boldsymbol{m}^l[t-1] + \frac{\boldsymbol{x}^{l-1}[t]}{t}, \quad (15)$$

$$\boldsymbol{x}^l[t] = t * (F^l(\boldsymbol{m}^l[t]) - F^l(\boldsymbol{m}^l[t-1])), \quad (16)$$

*where $\boldsymbol{m}^l[t]$ is the membrane potential at time-step $t$ which is equal to $\boldsymbol{b}^{l-1}$ if the previous layer has bias else $0$, $\boldsymbol{r}^l[t]$ is the encoded output activation value of the previous $t$ time-steps. The output of layer $l$ at time-step $t$, which serves as the input to layer $l+1$, is given by $\boldsymbol{x}^l[t]$.*

The proof of Theorem 4.2 is detailed in the Appendix C. From Theorem 4.2, a single-input unit requires two variables: one to record $\boldsymbol{m}^l[t]$ and another to record $F(\boldsymbol{m}^l[t])$, in order to reduce redundant calculations at each time-step.

**Theorem 4.3.** *Let $\cdot$ be an operation with two inputs, such as matrix multiplication or element-wise multiplication. In ANN-to-SNN conversion, the mapping from operation $\cdot$ to dynamics of the differential graded units in differential coding is given by Equations (17) to (19).*

$$\boldsymbol{m}_A^l[t] = \boldsymbol{r}_A^{l-1}[t] = \boldsymbol{m}_A^l[t-1] + \frac{\boldsymbol{x}_A^{l-1}[t]}{t}, \quad (17)$$

$$\boldsymbol{m}_B^l[t] = \boldsymbol{r}_B^{l-1}[t] = \boldsymbol{m}_B^l[t-1] + \frac{\boldsymbol{x}_B^{l-1}[t]}{t}, \quad (18)$$

$$\boldsymbol{x}^l[t] = \frac{\boldsymbol{x}_A^{l-1}[t] \cdot \boldsymbol{x}_B^{l-1}[t]}{t} + \boldsymbol{x}_A^{l-1}[t] \cdot \boldsymbol{m}_B^l[t] + \boldsymbol{m}_A^l[t] \cdot \boldsymbol{x}_B^{l-1}[t], \quad (19)$$

*where $\boldsymbol{m}_A^l[t]$ and $\boldsymbol{m}_B^l[t]$ are membrane potential at time-step $t$, and $\boldsymbol{r}_A^{l-1}[t]$ and $\boldsymbol{r}_B^{l-1}[t]$ are the encoded activation values of the previous layers at time-step $t$. The output of layer $l$ at time-step $t$, which serves as the input to layer $l+1$, is given by $\boldsymbol{x}^l[t]$.*

The proof of Theorem 4.3 is detailed in the Appendix D. From Theorem 4.3, a neuron with two inputs requires two variables to record $\boldsymbol{m}_A^l[t]$ and $\boldsymbol{m}_B^l[t]$, respectively. Graded units provide the ability to integrate information about nonlinear layer changes. This enables the conversion of various complex networks, including CNNs and Transformers.

### 4.1.2. DIFFERENTIAL SPIKING NEURONS

Since the majority of computations occur in fully connected layers, convolutional layers, and matrix multiplication layers, it is recommended to introduce spiking neuron layers before these layers, so that the computation is event-driven, thereby effectively reducing the network's energy consumption. Theorem 4.4 demonstrates how to convert a spiking

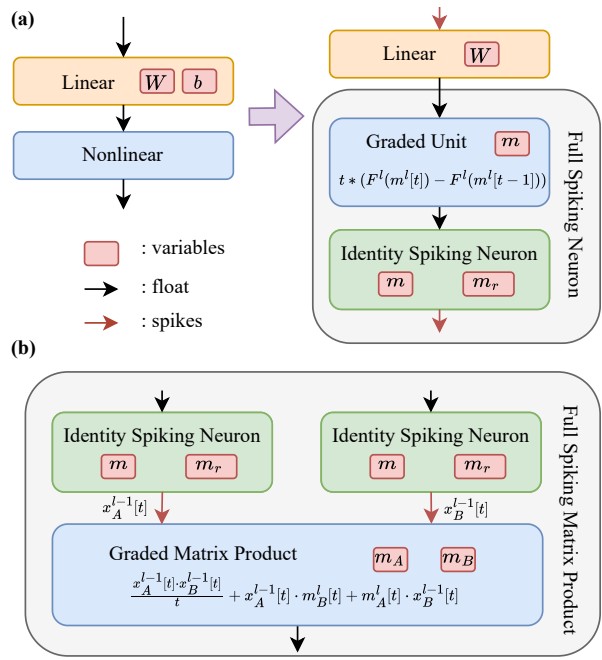

*Figure 3.* (a) Conversion of a linear layer followed by a nonlinear layer in an ANN into SNN modules. (b) Conversion of a matrix product or element-wise multiplication in the ANN into SNN modules.

neuron in rate coding into a differential neuron in differential coding.

**Theorem 4.4.** *In rate coding, the output of the previous layer, $\boldsymbol{x}^{l-1}[t]$, is directly used as the input current for the current layer $\boldsymbol{I}^l[t] = \boldsymbol{x}^{l-1}[t]$. In differential coding, the input current $\boldsymbol{I}^l[t]$ can be adjusted as shown in Equation (20), which converts any spiking neuron into a differential spiking neuron:*

$$\boldsymbol{I}^l[t] = \boldsymbol{m}_r^l[t] + \boldsymbol{x}^{l-1}[t], \quad (20)$$

$$\boldsymbol{m}_r^l[t+1] = \boldsymbol{m}_r^l[t] + \frac{\boldsymbol{x}^{l-1}[t]}{t} - \frac{\boldsymbol{x}^l[t]}{t}, \quad (21)$$

*where $\boldsymbol{m}_r^l[0]$ is $\boldsymbol{b}^{l-1}$ if the previous layer has bias else $0$.*

The proof of Theorem 4.4 is detailed in the Appendix E. In contrast to rate coding, which is constrained by a decay that limits the output range to below the threshold $\theta$, differential coding allows for adaptive adjustment of the neuron's output range by directly modifying the encoded activation $\boldsymbol{r}^l[t]$. This flexibility is especially beneficial in scenarios with multiple or dynamically adjustable thresholds, as the combination of different thresholds enhances the representation accuracy. So, we employ a differential version of identity multi-threshold spiking neuron in our experiments.

### 4.1.3. DIFFERENTIAL CODING FOR LINEAR LAYER

Theorem 4.5 shows the conversion of linear layers under differential coding in ANN-to-SNN conversion.

**Theorem 4.5.** *For linear layers, including fully connected and convolutional layers that can be represented by Equation (22),*

$$\boldsymbol{x}^l = \boldsymbol{W}^l \boldsymbol{x}^{l-1} + \boldsymbol{b}^l, \tag{22}$$

*where $\boldsymbol{W}^l$ and $\boldsymbol{b}^l$ is the weight and bias of layer $l$. Under differential coding in SNNs, this is equivalent to eliminating the bias term $\boldsymbol{b}^l$ and initializing the membrane potential of the subsequent layer with the bias value.*

The proof of Theorem 4.5 is detailed in the Appendix F. Figure 3 shows the overall method to replace ANN modules by SNN modules under differential coding.

### 4.2. Optimal Threshold for ReLU

When replacing the ReLU function in CNNs with spiking neurons, we propose an algorithm called threshold iteration method for determining the optimal threshold.

**Assumption 4.6.** According to (de G. Matthews et al., 2018), assume that the input $x$ to the neuron follows a normal distribution $X$ with mean $\mu$ and variance $\sigma^2$.

Based on Assumption 4.6, we introduce Definition 4.7 to define the overall error function, which is obtained by integrating the function error over the distribution of activation values.

**Definition 4.7.** In the $T$ time-steps conversion, the quantization and clipping errors of the ReLU function can be expressed as

$$QE(\theta) = \int_{-\infty}^{+\infty} \left(f(x,\theta) - \max(x,0)\right)^2 e^{-\frac{(x-\mu)^2}{2\sigma^2}} \mathrm{d}x, \tag{23}$$

$$f(x,\theta) = \frac{\theta}{N} clamp\left(\lfloor \frac{Nx + \frac{\theta}{2}}{\theta} \rfloor, 0, N\right), \tag{24}$$

where $f(x,\theta)$ represents the expected encoded activation in SNNs for a threshold $\theta$ which is proposed by (Bu et al., 2022a). For an IF neuron, $N = T$. For a multi-threshold with $n$ threshold, roughly let $N = 2^n T$.

Finding the optimal threshold by directly differentiating this function is challenging. However, we can take an alternative approach by introducing a variable $k$ to help determine the optimal threshold. We consider two cases: $k$ multiplies the output threshold amplitude as in Equation (25), and $k$ multiplies the threshold during spike calculation as in Equation (28). These cases yield the following two lemmas.

**Lemma 4.8.**

$$QE_1(\theta,k) = \int_{-\infty}^{+\infty} \left(f_1(x,\theta,k) - \max(x,0)\right)^2 e^{-\frac{(x-\mu)^2}{2\sigma^2}} dx, \tag{25}$$

$$f_1(x,\theta,k) = k\frac{\theta}{N} clamp\left(\lfloor \frac{Nx + \frac{\theta}{2}}{\theta} \rfloor, 0, N\right). \tag{26}$$

When $\theta$ is fixed, $QE_1(\theta, k)$ reaches its minimum value when:

$$k = k_1 = \frac{\mu}{\theta} \frac{1 - \sum_{i=1}^n \frac{1}{n} erf\left(\frac{\left(\frac{(2i-1)\theta}{2n} - \mu\right)}{\sqrt{2}\sigma}\right)}{1 - \sum_{i=1}^n \frac{2i-1}{n^2} erf\left(\frac{\left(\frac{(2i-1)\theta}{2n} - \mu\right)}{\sqrt{2}\sigma}\right)}$$

$$+ \frac{\sigma}{\sqrt{\frac{\pi}{2}}\theta} \frac{\sum_{i=1}^n \frac{1}{n} e^{-\frac{\left(\frac{(2i-1)\theta}{2n} - \mu\right)^2}{2\sigma^2}}}{1 - \sum_{i=1}^n \frac{2i-1}{n^2} erf\left(\frac{\left(\frac{(2i-1)\theta}{2n} - \mu\right)}{\sqrt{2}\sigma}\right)}. \tag{27}$$

**Lemma 4.9.**

$$QE_2(\theta,k) = \int_{-\infty}^{+\infty} \left(f_2(x,\theta,k) - \max(x,0)\right)^2 e^{-\frac{(x-\mu)^2}{2\sigma^2}} dx, \tag{28}$$

$$f_2(x,\theta,k) = \frac{\theta}{N} clamp\left(\lfloor \frac{Nx + \frac{k\theta}{2}}{k\theta} \rfloor, 0, N\right). \tag{29}$$

When $\theta$ is fixed, $QE_2(\theta, k)$ reaches its minimum value when $k = 1$.

---

**Algorithm 1** Threshold iteration method to find the best threshold

1: **Input:** Pre-trained ANN Model $F_{\text{ANN}}(\boldsymbol{W})$, Dataset $\boldsymbol{D}$.
2: **Initialize:** Set $\boldsymbol{\theta} \leftarrow 1$ (any positive initial value)
3: Run the model $F_{\text{ANN}}(\boldsymbol{W})$ on dataset $\boldsymbol{D}$ to statically compute the mean $\boldsymbol{\mu}$ and variance $\boldsymbol{\sigma}^2$ of pre-activations of each ReLU separately.
4: **repeat**
5:     Update $\boldsymbol{k_1}$ based on $\boldsymbol{\mu}$ and $\boldsymbol{\sigma}^2$ according to Eq (27)
6:     Update $\boldsymbol{\theta} \leftarrow \boldsymbol{k_1} \cdot \boldsymbol{\theta}$
7: **until** $1 - \epsilon < \boldsymbol{k_1} < 1 + \epsilon$, where $\epsilon$ tends to 0.
8: **Output:** Threshold $\boldsymbol{\theta}$

---

According to Lemma 4.8 and Lemma 4.9, we obtain the following inequality and Theorem 4.10:

$$QE(k_1\theta) < QE_2(k_1\theta, \tfrac{1}{k_1}) = QE_1(\theta, k_1) < QE(\theta). \tag{30}$$

**Theorem 4.10.** *Starting from any positive initial value of $\theta$, the rate of change $k_1$ can be continuously calculated based on the prior mean $\mu$, variance $\sigma^2$, and the current threshold $\theta$ using Equation (27). The iteration $\theta = k_1\theta$ continues until convergence, at which point the global optimal threshold $\theta$ is obtained. The process is guaranteed to converge as long as the threshold is greater than 0.*

The proof of Theorem 4.8, 4.9, and 4.10 are detailed in the Appendix G, H and I. Therefore, the optimal $\theta$ can be determined by the Theorem 4.10 and Algorithm 1.

### 4.3. Hardware implementation of MT Neuron

Equation (6) in Section 3.1 is presented for ease of understanding. In hardware implementation, the argmin module

is not used. We develop a hardware-friendly version of the MT neuron model, which can efficiently map the appropriate threshold using the potential's sign bit and exponent bits at an extremely low cost.

Compared with previous ANN2SNN methods, the MT neuron is required to transmit an extra index $i$ for the threshold. When implementing the MT neuron, two implementations can be considered:

1. Sent $V_{th}[i] \cdot S[t]$ to the next layer

2. Add an external threshold dimension with $2n$ elements to $S[t]$, set $S[t][i] = 1$ and $S[t][j] = 0$ for all $j \neq i$. At the same time, an external threshold dimension is added to the weight of the next layer, whose elements are the multi-level thresholds.

For simplicity, we use implementation 1 on GPUs, which is not pure binary but equivalent to implementation 2 with binary outputs. The MT neuron is also compatible with asynchronous computing neuromorphic chips because its outputs are still sparse events. Take the speck chip (Yao et al., 2024) as an example. The LIF neuron in the convolutional layer in speck chip outputs $(c, x, y)$ to the next layer. When using the MT neuron, the only modification is adding a threshold index, i.e., $(c, x, y, i)$. The computations of the next layer should also be changed by using a bit-shift operation on the weights, as the threshold is a power of 2 and this allows multiplication to be avoided. After the above modifications, the computation is still asynchronous and event-driven. The implementation to avoid argmin in Equation (6) in hardware can be described in the following two steps.

Step 1: Get SNN weights by using the weight normalization strategy (Rueckauer et al., 2017) described by the following equation.

$$W_{\text{SNN}}^l = W_{\text{ANN}}^l \frac{\theta^l}{\theta^{l+1}}, \tag{31}$$

$$b_{\text{SNN}}^l = \frac{b_{\text{ANN}}^l}{\theta^{l+1}}. \tag{32}$$

We then set all base thresholds $\theta^l = 1$, resulting in the following thresholds for the MT neuron:

$$\lambda_i^l = \begin{cases} \frac{1}{2^{i-1}}, & 1 < i \leq n, \\ \frac{-1}{2^{i-n-1}}, & n < i \leq 2n. \end{cases} \tag{33}$$

Step 2: We define $\frac{4}{3} m^l[t] = (-1)^S 2^E (1 + M)$ with 1 sign bit $(S)$, 8 exponent bits $(E)$, and 23 mantissa bits $(M)$. Since the median of $\frac{1}{2^{k-1}}$ and $\frac{1}{2^k}$ is $\frac{3}{4} \frac{1}{2^{k-1}}$, we can easily select the correct threshold index $i$ using $E$ and $S$ of $\frac{4}{3} m^l[t]$, without performing $2n$ subtractions to calculate the argmin in Equation (6):

$$\text{MTH}_{\theta,n}(m^l[t], i) = \begin{cases} 1, & \text{if } \begin{cases} i < n, \text{ S} = 0 \text{ and } i = 1 - \text{E}, \\ i \geq n, \text{ S} = 1 \text{ and } i - n = 1 - \text{E}, \end{cases} \\ 0, & \text{otherwise.} \end{cases} \tag{34}$$

For differential neurons, the memory overhead compared to initial neurons, such as IF or MT neurons, only includes an additional membrane potential. This extra potential is used to adjust the input current as described in Theorem 4.4.

To enable fast execution on GPU, we also design an efficient algorithm which is detailed in Appendix P.

## 5. Experimental Results

In this section, we first evaluate the performance of our proposed method on ImageNet dataset across different models, comparing our results with state-of-the-art ANN-to-SNN conversion methods. Then, we compute and analyze the energy consumption of the converted SNNs. Finally, we conduct comparative experiments to validate the effectiveness of differential coding and the threshold iteration method.

*Table 1.* Accuracy and energy ratio of DCGS(Ours) of different converted models on ImageNet Dataset

| Model Config | Time-step $T$ | | | | |
|---|---|---|---|---|---|
| Acc/Energy | 2 | 4 | 8 | 12 | 16 |
| ResNet34-4/1, Param:21.8M, Acc:76.42% | | | | | |
| Acc | 59.71 | 73.35 | 76.04 | 76.26 | 76.35 |
| Energy ratio | 0.14 | 0.24 | 0.37 | 0.46 | 0.53 |
| VGG16-4/1, Param:138M, Acc:73.25% | | | | | |
| Acc | 70.69 | 72.72 | 73.17 | 73.23 | 73.26 |
| Energy ratio | 0.10 | 0.15 | 0.22 | 0.26 | 0.29 |
| ViT-Small-8/4, Param:22.1M, Acc:81.38% | | | | | |
| Acc | 77.84 | 81.11 | 81.43 | 81.39 | 81.38 |
| Energy ratio | 0.32 | 0.62 | 1.05 | 1.39 | 1.71 |

### 5.1. Comparison with the State-of-the-art ANN-to-SNN Conversion Methods

We conducted conversion experiments on 11 different CNNs and Transformers using the Imagenet dataset. We denote the converted model as $model - n/c$, where the multi-threshold neurons have $n$ positive and $n$ opposing negative thresholds, and the calculated channel-wise thresholds are scaled by a factor $c$. Eg., the ResNet34-4/2 model represents the conversion using the ResNet34 model, employing multi-threshold spiking neurons with 4 positive and 4 negative thresholds, and the actual thresholds are based on the statistical thresholds multiplied by a factor of 2.

When $n = 1$, it can be treated as an IF neuron with an additional negative threshold. Table 2 shows a comparison

*Table 2.* Comparison between the proposed method and previous ANN-to-SNN conversion works on ImageNet dataset.

| Method | Type | Arch. | Param.(M) | ANN Acc(%) | $T$ | SNN Acc(%) |
|---|---|---|---|---|---|---|
| TS (Deng & Gu, 2021) | CNN-to-SNN | VGG-16 | 138 | 72.40 | 64 | 70.97 |
| SNM (Wang et al., 2022a) | CNN-to-SNN | VGG-16 | 138 | 73.18 | 64 | 71.50 |
| MMSE (Li et al., 2021) | CNN-to-SNN | ResNet-34 VGG-16 | 21.8 138 | 75.66 75.36 | 64 64 | 71.12 70.69 |
| QCFS (Bu et al., 2022b) | CNN-to-SNN | ResNet-34 VGG-16 | 21.8 138 | 74.32 74.29 | 64 64 | 72.35 72.85 |
| SRP (Hao et al., 2023a) | CNN-to-SNN | ResNet-34 VGG-16 | 21.8 138 | 74.32 74.29 | 4, 64 4, 64 | 66.71, 68.61 66.47, 69.43 |
| MST (Wang et al., 2023) | Transformer-to-SNN | Swin-T(BN) | 28.5 | 80.51 | 128, 512 | 77.88, 78.51 |
| STA (Jiang et al., 2024) | Transformer-to-SNN | ViT-B/32 | 86 | 83.60 | 32, 256 | 78.72, 82.79 |
| SpikeZIP-TF (kang you et al., 2024) | Transformer-to-SNN | SViT-S-32Level SViT-B-32Level SViT-L-32Level | 22.05 86.57 304.33 | 81.59 82.83 83.86 | 64 64 64 | 81.45 82.71 83.82 |
| ECMT (Huang et al., 2024) | Transformer-to-SNN | ViT-S/16 EVA-G | 22 1074 | 78.04 89.62 | 8, 10 4, 8 | 76.03, 77.07 88.60, 89.40 |
| **DCGS(Ours)** | CNN-to-SNN | ResNet18-1/1 ResNet34-1/1 VGG-1/1 ResNet18-4/1 ResNet34-4/1 VGG-4/1 | 11.7 21.8 138 11.7 21.8 138 | 71.49 76.42 73.25 71.49 76.42 73.25 | 32, 64 32, 64 32, 64 4, 8 4, 8 4, 8 | 69.89, 71.08 58.86, 74.11 72.04, 73.13 70.07, 71.31 73.35, 76.04 72.72, 73.17 |
| | Transformer-to-SNN | ViT-S-8/4 ViT-B-8/4 ViT-L-8/4 EVA02-T-8/4 EVA02-S-8/4 EVA02-B-8/4 EVA02-L-8/4 | 22.1 86.6 304.3 5.8 22.1 87.1 305.1 | 81.38 84.54 85.84 80.63 85.73 88.69 90.05 | 2, 4 2, 4 2, 4 2, 4 2, 4 2, 4 2, 4 | 77.84, 81.11 80.34, 83.98 83.73, 85.45 66.32, 79.56 71.37, 84.70 84.62, 88.16 88.25, 89.72 |

of our method with other ANN-to-SNN conversion methods, and detailed results can be found in Appendix K.

In CNNs, when $n = 1$, our method outperforms the existing methods on the same structure achieving state-of-the-art results; and when $n > 1$, we achieve better performance with extremely shorter time-steps.

In Transformers, the threshold iteration method is not suitable, and using the top 99.9% of activation values does not optimal thresholds. As a result, achieving high performance with $n = 1$ in short time-steps is challenging. Therefore, we scale the statistical thresholds by $c = 4$ and setting $n = 8$. Our method requires no training and achieves high performance in extremely short time-steps.

### 5.2. Energy Estimation and Result Analysis

Based on (Horowitz, 2014), we use Equation (35) to estimate the energy consumption ratio of the converted SNN relative to the ANN, with $E_{\text{MAC}} = 4.6\text{pJ}$ and $E_{\text{AC}} = 0.9\text{pJ}$.

$$\frac{E_{\text{SNN}}}{E_{\text{ANN}}} = \frac{MACs_{\text{SNN}} * E_{\text{MAC}} + ACs_{\text{SNN}} * E_{\text{AC}}}{MACs_{\text{ANN}} * E_{\text{MAC}}}. \quad (35)$$

Since most computations in the network occur in the fully connected, convolutional, and matrix multiplication layers, which in SNNs are primarily implemented by additions (with $ACs_{\text{SNN}} >> MACs_{\text{SNN}}$), we approximate $MACs_{\text{SNN}} \approx 0$. We then use the statistical spike emission rate $\eta$ to estimate $\frac{ACs_{\text{SNN}}}{MACs_{\text{ANN}}}$, thereby estimating the energy consumption of the SNN relative to the pre-conversion ANN. Table 1 presents partial results, and the detailed results for all converted SNN models can be found in Appendix K.

For CNNs, our method achieves SNN performance comparable to the ANN with low power consumption and extremely short time-steps. Notably, for the VGG16 model, it achieves an accuracy of 73.17% with only a 0.08% accuracy loss and 22% power consumption.

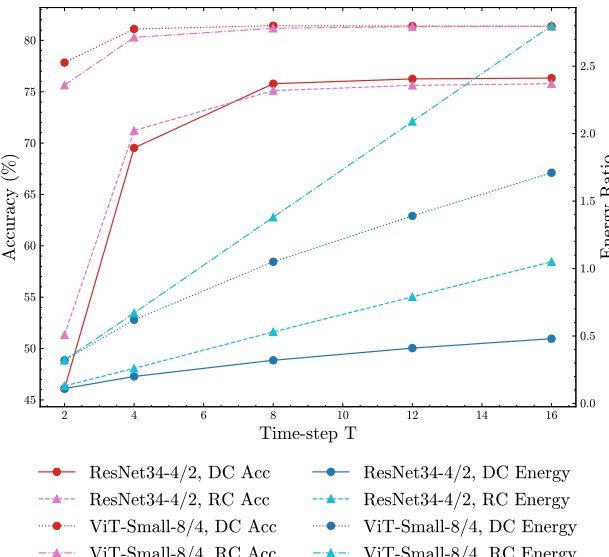

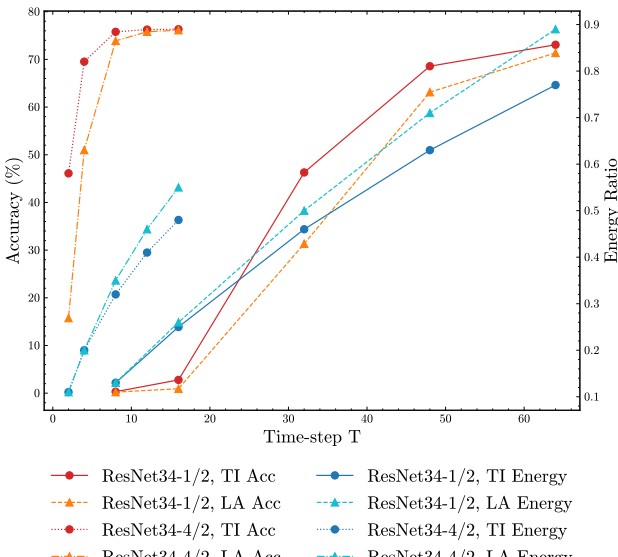

*Figure 4.* Effective of the differential coding compared to the rate coding. DC and RC represent Differential Coding and Rate Coding, respectively.

*Figure 5.* Effectiveness of the threshold iteration method. TI and LA represent Threshold Iteration Method and 99.9% large activation method respectively.

For Transformers, although our method achieves high accuracy with extremely short time-steps and shows a decreasing energy consumption growth rate, there is still significant room for further optimization. This is primarily due to the lack of an optimal threshold calculation method, which causes inefficient spike firing in the SNNs. This leads to larger errors when matching the ANN activation values, resulting in more premature spike emissions. This is an area we aim to improve in future research.

### 5.3. Effectiveness of the Differential Coding

To validate the effectiveness of the differential coding, we compared the performance of differential coding and rate coding using the same model. The partial visualization results are presented in Figure 4, with a more detailed table provided in Appendix L. The model using differential coding not only outperforms the rate coding model in terms of accuracy, but also consumes less energy. This is because differential coding directly updates the current encoding value based on previous results, avoiding decay. It can represent a broader range and steadily improve representation accuracy. Once the representation precision reaches a certain level, no further spikes are emitted.

### 5.4. Effectiveness of Threshold Iteration Method

To verify the effectiveness of the threshold iteration method, we compared the performance of the converted SNNs using two different methods, threshold iteration method and

the top 99.9% of activation method, with different numbers of threshold neurons in ResNet34. Here we set scale factor $c = 2$ to prevent the accuracy from being too small when using the 99.9% large activation method. The partial visualization results are presented in Figure 5, and more information can be found in Appendix M. The experimental results show that the thresholds derived using the threshold iteration method outperform those obtained through the 99.9% large activation method, achieving better accuracy and lower energy consumption at each time-step.

## 6. Conclusion

This article introduces a training-free ANN-to-SNN conversion method based on differential coding. Instead of directly encoding rate information, it uses spikes to encode differential information, improving both network accuracy and energy efficiency. For ReLU conversions, it includes a threshold iteration method to find the optimal thresholds, which further enhances the network performance.

However, the proposed method also has some limitations. Differential coding requires spiking neurons to have at least one negative threshold to generate negative spikes for error correction; otherwise, excessive spike errors will accumulate continuously. Meanwhile, we have not develop a method to determine the optimal thresholds for Transformers, which limits the conversion performance on Transformers. Future research could focus on addressing this challenge.

## Acknowledgments

This work was supported by STI 2030-Major Projects 2021ZD0200300, the National Natural Science Foundation of China (62422601, U24B20140, and 62088102), Beijing Municipal Science and Technology Program (Z241100004224004), Beijing Nova Program (20230484362, 20240484703), and National Key Laboratory for Multimedia Information Processing.

## Impact Statement

This paper presents work whose goal is to advance the field of Machine Learning. There are many potential societal consequences of our work, none which we feel must be specifically highlighted here.

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

## A. Overall Algorithm

Algorithm 2 outlines the whole procedures we adopt.

---

**Algorithm 2** Differential Coding with Graded Units and Spiking Neurons (DCGS) Conversion Method

1: **Input:** Pre-trained ANN model $F_{\text{ANN}}(\boldsymbol{W})$, Dataset $\boldsymbol{D}$. Time-step $T$, or Threshold percentage $p$ and scaling factor $c$.
2: **Output:** Converted SNN model $F_{\text{SNN}}(\boldsymbol{W}, \boldsymbol{\theta}, \boldsymbol{v})$
3: **Step 1: Determine the Threshold:**
4: **if** $F_{\text{ANN}}(\boldsymbol{W})$ is a ReLU network **then**
5:    Use the threshold iteration method with $T$ to calculate threshold $\boldsymbol{\theta}$ on dataset $\boldsymbol{D}$
6: **else**
7:    Static threshold $\boldsymbol{\theta}$ as the top $p\%$ of activation values on dataset $\boldsymbol{D}$, and multiply by the scaling factor $c$
8: **end if**
9: **Step 2: Replace Modules:**
10: Replace the nonlinear layer with a differential graded unit.
11: Insert a differential identity spiking neuron before each linear layer.
12: Remove the bias $\boldsymbol{b}$ from the linear layer and set the initial potential $\boldsymbol{v} = \boldsymbol{b}$ for the next layer.
13: **Return** the converted SNN model $F_{\text{SNN}}(\boldsymbol{W}, \boldsymbol{\theta}, \boldsymbol{v})$

---

## B. Explanation of Definition 4.1

**Definition B.1. (Repeated from Definition 4.1)** In differential coding, the encoded activation value $\boldsymbol{r}^l[t]$ is defined as shown in Equation (14), where $\boldsymbol{e}^l[t]$ represents the encoded output value of the neuron at time-step $t$, and $\boldsymbol{x}^l[t]$ represents the actual output value of the neuron. The relationship between the two is expressed by Equation (13), as follows:

$$\boldsymbol{e}^l[t] = \boldsymbol{r}^l[t-1] + \boldsymbol{x}^l[t], \tag{36}$$

$$\boldsymbol{r}^l[t] = \boldsymbol{r}^l[t-1] + \frac{\boldsymbol{x}^l[t]}{t} = \frac{1}{t}\sum_{i=1}^{t} \boldsymbol{e}^l[i], \tag{37}$$

where $t$ starts from 1, $\boldsymbol{r}^l[0] = 0$.

*Proof.* In this definition, $\boldsymbol{e}^l[t]$ is essentially adjusted based on the historical encoded values. If no spike is emitted, then $\boldsymbol{e}^l[t] = \boldsymbol{r}^l[t-1]$, also ensuring that the encoded value $\boldsymbol{r}^l[t] = \boldsymbol{r}^l[t-1]$. The derivation of Equation (37) can be written as:

$$\begin{aligned}
\boldsymbol{r}^l[t] &= \boldsymbol{r}^l[t-1] + \frac{\boldsymbol{x}^l[t]}{t} \\
&= \frac{\boldsymbol{r}^l[t-1]}{t} + \frac{\boldsymbol{x}^l[t]}{t} + \frac{t-1}{t}\boldsymbol{r}^l[t-1] \\
&= \frac{\boldsymbol{e}^l[t]}{t} + \frac{t-1}{t}\boldsymbol{r}^l[t-1] \\
&= \frac{\boldsymbol{e}^l[t]}{t} + \frac{\boldsymbol{e}^l[t-1]}{t-1} + \frac{t-2}{t-1}\boldsymbol{r}^l[t-2] \\
&= ... \\
&= \frac{1}{t}\sum_{i=1}^{t}\boldsymbol{e}^l[i] + \frac{0}{1}\boldsymbol{r}^l[0] \\
&= \frac{1}{t}\sum_{i=1}^{t}\boldsymbol{e}^l[i]
\end{aligned} \tag{38}$$

$\square$

## C. Proof of Theorem 4.2

**Theorem C.1.** *(Repeated from Theorem 4.2) Let $F^l$ be a nonlinear layer $l$ with only one input $\boldsymbol{x}^{l-1}[t]$, such as Gelu, Silu, Maxpool, LayerNorm, or Softmax. In ANN-to-SNN conversion, the mapping from $F$ to dynamics of the differential graded unit in differential coding is given by Equations (15) and (16).*

$$\boldsymbol{m}^l[t] = \boldsymbol{r}^{l-1}[t] = \boldsymbol{m}^l[t-1] + \frac{\boldsymbol{x}^{l-1}[t]}{t}, \tag{39}$$

$$\boldsymbol{x}^l[t] = t * (F^l(\boldsymbol{m}^l[t]) - F^l(\boldsymbol{m}^l[t-1])), \tag{40}$$

*where $\boldsymbol{m}^l[t]$ is the membrane potential at time-step $t$ which is equal to the encoded input value, $\boldsymbol{r}^l[t]$ is the encoded output activation value of the previous $t$ time-steps. The output of layer $l$ at time-step $t$, which serves as the input to layer $l+1$, is given by $\boldsymbol{x}^l[t]$.*

*Proof.*

$$\boldsymbol{m}^l[t] = \boldsymbol{r}^{l-1}[t] = \boldsymbol{r}^{l-1}[t-1] + \frac{\boldsymbol{x}^{l-1}[t]}{t} = \boldsymbol{m}^l[t-1] + \frac{\boldsymbol{x}^{l-1}[t]}{t} \tag{41}$$

$$\boldsymbol{x}^l[t] = t * (\boldsymbol{r}^l[t] - \boldsymbol{r}^l[t-1]) = t * (F(\boldsymbol{r}^{l-1}[t]) - F(\boldsymbol{r}^{l-1}[t-1])) = t * (F(\boldsymbol{m}^l[t]) - F(\boldsymbol{m}^l[t-1])) \tag{42}$$

$\square$

From Theorem 4.2, a single-input unit requires two variables: one to record $\boldsymbol{m}^l[t]$ and another to record $F(\boldsymbol{m}^l[t])$, in order to reduce redundant calculations at each time-step.

## D. Proof of Theorem 4.3

**Theorem D.1.** *(Repeated from Theorem 4.3) Let $\cdot$ be an operation with two inputs, such as matrix multiplication or element-wise multiplication. In ANN-to-SNN conversion, the mapping from operation $\cdot$ to dynamics of the differential graded unit in differential coding is given by Equations (43) to (45).*

$$\boldsymbol{m}_A^l[t] = \boldsymbol{r}_A^{l-1}[t] = \boldsymbol{m}_A^l[t-1] + \frac{\boldsymbol{x}_A^{l-1}[t]}{t}, \tag{43}$$

$$\boldsymbol{m}_B^l[t] = \boldsymbol{r}_B^{l-1}[t] = \boldsymbol{m}_B^l[t-1] + \frac{\boldsymbol{x}_B^{l-1}[t]}{t}, \tag{44}$$

$$\boldsymbol{x}^l[t] = \frac{\boldsymbol{x}_A^{l-1}[t] \cdot \boldsymbol{x}_B^{l-1}[t]}{t} + \boldsymbol{x}_A^{l-1}[t] \cdot \boldsymbol{m}_B^l[t] + \boldsymbol{m}_A^l[t] \cdot \boldsymbol{x}_B^{l-1}[t], \tag{45}$$

*where $\boldsymbol{m}_A^l[t]$ and $\boldsymbol{m}_B^l[t]$ are membrane potential at time-step $t$, and $\boldsymbol{r}_A^{l-1}[t]$ and $\boldsymbol{r}_B^{l-1}[t]$ are the encoded activation values of the previous layers at time-step $t$. The output of layer $l$ at time-step $t$, which serves as the input to layer $l+1$, is given by $\boldsymbol{x}^l[t]$.*

*Proof.*

$$\boldsymbol{m}_A^l[t] = \boldsymbol{r}_A^{l-1}[t] = \boldsymbol{r}_A^{l-1}[t-1] + \frac{\boldsymbol{x}_A^{l-1}[t]}{t} = \boldsymbol{m}_A^l[t-1] + \frac{\boldsymbol{x}_A^{l-1}[t]}{t} \tag{46}$$

$$\boldsymbol{m}_B^l[t] = \boldsymbol{r}_B^{l-1}[t] = \boldsymbol{r}_B^{l-1}[t-1] + \frac{\boldsymbol{x}_B^{l-1}[t]}{t} = \boldsymbol{m}_B^l[t-1] + \frac{\boldsymbol{x}_B^{l-1}[t]}{t} \tag{47}$$

$$
\begin{aligned}
\boldsymbol{x}^l[t] &= t * (\boldsymbol{r}^l[t] - \boldsymbol{r}^l[t-1]) \\
&= t * (\boldsymbol{m}_A^l[t] \cdot \boldsymbol{m}_B^l[t] - \boldsymbol{m}_A^l[t-1] \cdot \boldsymbol{m}_B^l[t-1]) \\
&= t * ((\boldsymbol{m}_A^l[t-1] + \frac{\boldsymbol{x}_A^{l-1}[t]}{t}) \cdot (\boldsymbol{m}_B^l[t-1] + \frac{\boldsymbol{x}_B^{l-1}[t]}{t}) - \boldsymbol{m}_A^l[t-1] \cdot \boldsymbol{m}_B^l[t-1]) \\
&= \frac{\boldsymbol{x}_A^{l-1}[t] \cdot \boldsymbol{x}_B^{l-1}[t]}{t} + \boldsymbol{x}_A^{l-1}[t] \cdot \boldsymbol{m}_B^l[t-1] + \boldsymbol{m}_A^l[t-1] \cdot \boldsymbol{x}_B^{l-1}[t]
\end{aligned}
\tag{48}
$$

$\square$

From Theorem 4.3, a neuron with two inputs requires two variables to record $m_A^l[t]$ and $m_B^l[t]$, respectively.

## E. Proof of Theorem 4.4

**Theorem E.1.** *(Repeated from Theorem 4.4) In rate coding, the output of the previous layer, $x^{l-1}[t]$, is directly used as the input current for the current layer $I^l[t] = x^{l-1}[t]$. In differential coding, the input current $I^l[t]$ can be adjusted as shown in Equation (49), which converts any spiking neuron into a differential spiking neuron:*

$$I^l[t] = m_r^l[t] + x^{l-1}[t], \tag{49}$$

$$m_r^l[t+1] = m_r^l[t] + \frac{x^{l-1}[t]}{t} - \frac{x^l[t]}{t}, \tag{50}$$

*where $m_r^l[0]$ is $b^{l-1}$ if the previous layer has bias else $0$.*

*Proof.* Let the expected input encoding value of the differential neuron at the $l$-th layer at time-step $t$ be $r^{l-1}[t]$, and the expected output encoding value be $r^l[t]$. Due to soft resetting, the total expected membrane potential change is $m_r^l[t] = r^{l-1}[t-1] - r^l[t-1]$. The total input current is then:

$$I^l[t] = r^{l-1}[t-1] - r^l[t-1] + x^{l-1}[t] = m_r^l[t] + x^{l-1}[t] \tag{51}$$

Since by Definition 4.1:

$$r^{l-1}[t] = r^{l-1}[t-1] + \frac{x^{l-1}[t]}{t} \tag{52}$$

$$r^l[t] = r^l[t-1] + \frac{x^l[t]}{t} \tag{53}$$

we have:

$$m_r^l[t+1] = r^{l-1}[t] - r^l[t] \tag{54}$$

$$= r^{l-1}[t-1] + \frac{x^{l-1}[t]}{t} - r^l[t-1] - \frac{x^l[t]}{t} \tag{55}$$

$$= m_r^l[t] + \frac{x^{l-1}[t]}{t} - \frac{x^l[t]}{t}. \tag{56}$$

$\square$

## F. Proof of Theorem 4.5

**Theorem F.1.** *(Repeated from Theorem 4.5) For linear layers, including linear and convolutional layers that can be represented by Equation (57),*

$$x^l = W^l x^{l-1} + b^l, \tag{57}$$

*where $W^l$ and $b^l$ is the weight and bias of layer $l$.*

*Under differential coding, this is equivalent to eliminating the bias term $b^l$ by initializing the membrane potential of the subsequent layer with the bias value or adding the bias to the first input current of that layer, and then running the dynamic process at each time-step according to the following equation:*

$$x^l[t] = W^l x^{l-1}[t] \tag{58}$$

*Proof.* In the context of ANN-to-SNN conversion using rate coding, the output $x^l[t]$ of layer $l$ can be expressed as:

$$x^l[t] = W^l x^{l-1}[t] + b^l, \tag{59}$$

Under differential coding as defined in Definition B.1, we have:

$$e^l[t] = r^l[t-1] + x^l[t], \tag{60}$$

$$r^l[t] = r^l[t-1] + \frac{x^l[t]}{t} \tag{61}$$

$$r^l[0] = 0 \tag{62}$$

$$e^l[t] = W^l e^{l-1}[t] + b^l \tag{63}$$

$$r^l[t] = W^l r^{l-1}[t] + b^l \tag{64}$$

For $t > 1$, the following transformation holds:

$$
\begin{aligned}
x^l[t] = e^l[t] - r^l[t-1] &= t * r^l[t] - (t-1) * r^l[t-1] - r^l[t-1] \\
&= t * (W^l r^{l-1}[t] + b^l - W^l r^{l-1}[t-1] - b^l) \\
&= t * W^l \frac{1}{t} x^{l-1}[t] \\
&= W^l x^{l-1}[t]
\end{aligned}
\tag{65}
$$

when $t = 1$, we can let $r^l[0] = b^l$, and initialize the membrane potential of the subsequent layer with the bias $b^l$ or adding the bias to the first input current of that layer and start running from time-step 1:

$$
\begin{aligned}
x^l[t] = e^l[t] - r^l[t-1] &= e^l[1] - r^l[0] \\
&= W^l e^{l-1}[t] + b^l - b^l = W^l x^{l-1}[t] + W^l r^{l-1}[0] \\
&= W^l x^{l-1}[t]
\end{aligned}
\tag{66}
$$

Therefore, for any $t > 0$ in differential coding, we have:

$$x^l[t] = W^l x^{l-1}[t] \tag{67}$$

$\square$

# G. Proof of Lemma 4.8

We first prove a previous lemma before proving Lemma 4.8.

**Lemma G.1.**

$$
\begin{aligned}
\int_a^b (c-x)^2 e^{-\frac{(x-\mu)^2}{2\sigma^2}} dx =& \sqrt{\tfrac{\pi}{2}} \sigma \left( \sigma^2 + \mu^2 - 2c\mu + c^2 \right) \left( erf\left( \frac{(b-\mu)}{\sqrt{2}\sigma} \right) - erf\left( \frac{(a-\mu)}{\sqrt{2}\sigma} \right) \right) \\
&+ \left( -\sigma^2 (b + \mu - 2c) \right) e^{-\frac{(b-\mu)^2}{2\sigma^2}} + \left( \sigma^2 (a + \mu - 2c) \right) e^{-\frac{(a-\mu)^2}{2\sigma^2}}
\end{aligned}
\tag{68}
$$

*Proof.* We first calculate $\int_a^b e^{-\frac{(x-\mu)^2}{2\sigma^2}} dx$, $\int_a^b x e^{-\frac{(x-\mu)^2}{2\sigma^2}} dx$ and $\int_a^b x^2 e^{-\frac{(x-\mu)^2}{2\sigma^2}} dx$ separately. Since $\text{erf}(x) = \frac{2}{\sqrt{\pi}} \int_0^x e^{-t^2} dt$, $\int_0^x e^{-t^2} dt = \frac{\sqrt{\pi}}{2} \text{erf}(x)$

$$
\begin{aligned}
\int_a^b e^{-\frac{(x-\mu)^2}{2\sigma^2}} dx &= \sqrt{2}\sigma \int_a^b e^{-\frac{(x-\mu)^2}{2\sigma^2}} d\frac{(x-\mu)}{\sqrt{2}\sigma} = \sqrt{2}\sigma \frac{\sqrt{\pi}}{2} \left( \text{erf}\left( \frac{(b-\mu)}{\sqrt{2}\sigma} \right) - \text{erf}\left( \frac{(a-\mu)}{\sqrt{2}\sigma} \right) \right) \\
&= \sqrt{\tfrac{\pi}{2}} \sigma \left( \text{erf}\left( \frac{(b-\mu)}{\sqrt{2}\sigma} \right) - \text{erf}\left( \frac{(a-\mu)}{\sqrt{2}\sigma} \right) \right)
\end{aligned}
\tag{69}
$$

$$
\begin{aligned}
\int_a^b x e^{-\frac{(x-\mu)^2}{2\sigma^2}} dx &= \int_a^b (x - \mu + \mu) e^{-\frac{(x-\mu)^2}{2\sigma^2}} dx = -\sigma^2 \int_a^b e^{-\frac{(x-\mu)^2}{2\sigma^2}} d\left( -\frac{(x-\mu)^2}{2\sigma^2} \right) + \mu \int_a^b e^{-\frac{(x-\mu)^2}{2\sigma^2}} dx \\
&= -\sigma^2 \left( e^{-\frac{(b-\mu)^2}{2\sigma^2}} - e^{-\frac{(a-\mu)^2}{2\sigma^2}} \right) + \mu \sqrt{\tfrac{\pi}{2}} \sigma \left( \text{erf}\left( \frac{(b-\mu)}{\sqrt{2}\sigma} \right) - \text{erf}\left( \frac{(a-\mu)}{\sqrt{2}\sigma} \right) \right)
\end{aligned}
\tag{70}
$$

$$\int_a^b x^2 e^{-\frac{(x-\mu)^2}{2\sigma^2}} \, \mathrm{d}x = \int_a^b \left( (x-\mu)^2 + 2\mu(x-\mu) + \mu^2 \right) e^{-\frac{(x-\mu)^2}{2\sigma^2}} \, \mathrm{d}x$$

$$= -\sigma^2 \int_a^b (x-\mu)\, de^{-\frac{(x-\mu)^2}{2\sigma^2}} + \int_a^b \left( 2\mu(x-\mu) + \mu^2 \right) e^{-\frac{(x-\mu)^2}{2\sigma^2}} \, \mathrm{d}x$$

$$= -\sigma^2 (b-\mu) e^{-\frac{(b-\mu)^2}{2\sigma^2}} + \sigma^2 (a-\mu) e^{-\frac{(a-\mu)^2}{2\sigma^2}} + \sigma^2 \int_a^b e^{-\frac{(x-\mu)^2}{2\sigma^2}} \, d(x-\mu)$$

$$- 2\sigma^2 \mu \left( e^{-\frac{(b-\mu)^2}{2\sigma^2}} - e^{-\frac{(a-\mu)^2}{2\sigma^2}} \right) + \mu^2 \sqrt{\frac{\pi}{2}} \sigma \left( \mathrm{erf}\left( \frac{(b-\mu)}{\sqrt{2}\sigma} \right) - \mathrm{erf}\left( \frac{(a-\mu)}{\sqrt{2}\sigma} \right) \right) \tag{71}$$

$$= -\sigma^2 (b-\mu) e^{-\frac{(b-\mu)^2}{2\sigma^2}} + \sigma^2 (a-\mu) e^{-\frac{(a-\mu)^2}{2\sigma^2}} + \sigma^2 \sqrt{\frac{\pi}{2}} \sigma \left( \mathrm{erf}\left( \frac{(b-\mu)}{\sqrt{2}\sigma} \right) - \mathrm{erf}\left( \frac{(a-\mu)}{\sqrt{2}\sigma} \right) \right)$$

$$- 2\sigma^2 \mu \left( e^{-\frac{(b-\mu)^2}{2\sigma^2}} - e^{-\frac{(a-\mu)^2}{2\sigma^2}} \right) + \mu^2 \sqrt{\frac{\pi}{2}} \sigma \left( \mathrm{erf}\left( \frac{(b-\mu)}{\sqrt{2}\sigma} \right) - \mathrm{erf}\left( \frac{(a-\mu)}{\sqrt{2}\sigma} \right) \right)$$

$$= \left( -\sigma^2 (b+\mu) \right) e^{-\frac{(b-\mu)^2}{2\sigma^2}} + \left( \sigma^2 (a+\mu) \right) e^{-\frac{(a-\mu)^2}{2\sigma^2}} + \sqrt{\frac{\pi}{2}} \sigma \left( \sigma^2 + \mu^2 \right) \left( \mathrm{erf}\left( \frac{(b-\mu)}{\sqrt{2}\sigma} \right) - \mathrm{erf}\left( \frac{(a-\mu)}{\sqrt{2}\sigma} \right) \right)$$

Finally, aggregate and calculate the answer:

$$\int_a^b (c-x)^2 e^{-\frac{(x-\mu)^2}{2\sigma^2}} \, \mathrm{d}x = \int_a^b \left( x^2 - 2cx + c^2 \right) e^{-\frac{(x-\mu)^2}{2\sigma^2}} \, \mathrm{d}x$$

$$= \left( -\sigma^2 (b+\mu) \right) e^{-\frac{(b-\mu)^2}{2\sigma^2}} + \left( \sigma^2 (a+\mu) \right) e^{-\frac{(a-\mu)^2}{2\sigma^2}} + \sqrt{\frac{\pi}{2}} \sigma \left( \sigma^2 + \mu^2 \right) \left( \mathrm{erf}\left( \frac{(b-\mu)}{\sqrt{2}\sigma} \right) - \mathrm{erf}\left( \frac{(a-\mu)}{\sqrt{2}\sigma} \right) \right)$$

$$- 2c \left( -\sigma^2 \left( e^{-\frac{(b-\mu)^2}{2\sigma^2}} - e^{-\frac{(a-\mu)^2}{2\sigma^2}} \right) + \mu \sqrt{\frac{\pi}{2}} \sigma \left( \mathrm{erf}\left( \frac{(b-\mu)}{\sqrt{2}\sigma} \right) - \mathrm{erf}\left( \frac{(a-\mu)}{\sqrt{2}\sigma} \right) \right) \right) \tag{72}$$

$$+ c^2 \sqrt{\frac{\pi}{2}} \sigma \left( \mathrm{erf}\left( \frac{(b-\mu)}{\sqrt{2}\sigma} \right) - \mathrm{erf}\left( \frac{(a-\mu)}{\sqrt{2}\sigma} \right) \right)$$

$$= \sqrt{\frac{\pi}{2}} \sigma \left( \sigma^2 + \mu^2 - 2c\mu + c^2 \right) \left( \mathrm{erf}\left( \frac{(b-\mu)}{\sqrt{2}\sigma} \right) - \mathrm{erf}\left( \frac{(a-\mu)}{\sqrt{2}\sigma} \right) \right)$$

$$+ \left( -\sigma^2 (b+\mu-2c) \right) e^{-\frac{(b-\mu)^2}{2\sigma^2}} + \left( \sigma^2 (a+\mu-2c) \right) e^{-\frac{(a-\mu)^2}{2\sigma^2}}$$

$$\square$$

Then, we can prove Lemma 4.8 base on Lemma G.1.

**Lemma G.2.** *(Repeated from Lemma 4.8)*

$$QE_1(\theta, k) = \int_{-\infty}^{+\infty} \left(f_1(x, \theta, k) - \max(x, 0)\right)^2 e^{-\frac{(x-\mu)^2}{2\sigma^2}} dx \tag{73}$$

$$f_1(x, \theta, k) = k\frac{\theta}{N} clamp\left(\lfloor\frac{Nx + \frac{\theta}{2}}{\theta}\rfloor, 0, N\right) \tag{74}$$

*When $\theta$ is fixed, $QE_1(\theta, k)$ reaches its minimum value when:*

$$k = k_1 = \frac{\mu}{\theta} \frac{1 - \sum_{i=1}^{n} \frac{1}{n} erf\left(\frac{\left(\frac{(2i-1)\theta}{2n} - \mu\right)}{\sqrt{2}\sigma}\right)}{1 - \sum_{i=1}^{n} \frac{2i-1}{n^2} erf\left(\frac{\left(\frac{(2i-1)\theta}{2n} - \mu\right)}{\sqrt{2}\sigma}\right)} + \frac{\sigma}{\sqrt{\frac{\pi}{2}}\theta} \frac{\sum_{i=1}^{n} \frac{1}{n} e^{-\frac{\left(\frac{(2i-1)\theta}{2n} - \mu\right)^2}{2\sigma^2}}}{1 - \sum_{i=1}^{n} \frac{2i-1}{n^2} erf\left(\frac{\left(\frac{(2i-1)\theta}{2n} - \mu\right)}{\sqrt{2}\sigma}\right)} \tag{75}$$

*Proof.* According to Lemma G.1, we have:

$$\int_a^b (c - x)^2 e^{-\frac{(x-\mu)^2}{2\sigma^2}} dx = \sqrt{\frac{\pi}{2}}\sigma\left(\sigma^2 + \mu^2 - 2c\mu + c^2\right)\left(erf\left(\frac{(b-\mu)}{\sqrt{2}\sigma}\right) - erf\left(\frac{(a-\mu)}{\sqrt{2}\sigma}\right)\right)$$
$$+ \left(-\sigma^2(b + \mu - 2c)\right) e^{-\frac{(b-\mu)^2}{2\sigma^2}} + \left(\sigma^2(a + \mu - 2c)\right) e^{-\frac{(a-\mu)^2}{2\sigma^2}} \tag{76}$$

Expand the calculation of $QE_1$:

$$\int_{-\infty}^{+\infty} \left(k\frac{\theta}{n} clamp\left(\lfloor\frac{nx + \frac{\theta}{2}}{\theta}\rfloor, 0, n\right) - \max(x, 0)\right)^2 e^{-\frac{(x-\mu)^2}{2\sigma^2}} dx$$

$$= \int_0^{+\infty} \left(k\frac{\theta}{n} clamp\left(\lfloor\frac{nx + \frac{\theta}{2}}{\theta}\rfloor, 0, n\right) - x\right)^2 e^{-\frac{(x-\mu)^2}{2\sigma^2}} dx$$

$$= \int_0^{\frac{\theta}{2n}} (x)^2 e^{-\frac{(x-\mu)^2}{2\sigma^2}} dx + \sum_{i=1}^{n-1} \int_{\frac{(2i-1)\theta}{2n}}^{\frac{(2i+1)\theta}{2n}} \left(k\frac{i\theta}{n} - x\right)^2 e^{-\frac{(x-\mu)^2}{2\sigma^2}} dx + \int_{\frac{(2n-1)\theta}{2n}}^{+\infty} (k\theta - x)^2 e^{-\frac{(x-\mu)^2}{2\sigma^2}} dx$$

$$= \sqrt{\frac{\pi}{2}}\sigma\left(\sigma^2 + \mu^2\right)\left(erf\left(\frac{\left(\frac{\theta}{2n} - \mu\right)}{\sqrt{2}\sigma}\right) - erf\left(\frac{-\mu}{\sqrt{2}\sigma}\right)\right) + \left(-\sigma^2\left(\frac{\theta}{2n} + \mu\right)\right) e^{-\frac{\left(\frac{\theta}{2n} - \mu\right)^2}{2\sigma^2}} + \left(\sigma^2\mu\right) e^{-\frac{\mu^2}{2\sigma^2}} \tag{77}$$

$$+ \sum_{i=1}^{n-1} \sqrt{\frac{\pi}{2}}\sigma\left(\sigma^2 + \mu^2 - 2k\frac{i\theta}{n}\mu + \left(k\frac{i\theta}{n}\right)^2\right)\left(erf\left(\frac{\left(\frac{(2i+1)\theta}{2n} - \mu\right)}{\sqrt{2}\sigma}\right) - erf\left(\frac{\left(\frac{(2i-1)\theta}{2n} - \mu\right)}{\sqrt{2}\sigma}\right)\right)$$

$$+ \sum_{i=1}^{n-1} \left(-\sigma^2\left(\frac{(2i+1)\theta}{2n} + \mu - 2k\frac{i\theta}{n}\right)\right) e^{-\frac{\left(\frac{(2i+1)\theta}{2n} - \mu\right)^2}{2\sigma^2}} + \sum_{i=1}^{n-1} \left(\sigma^2\left(\frac{(2i-1)\theta}{2n} + \mu - 2k\frac{i\theta}{n}\right)\right) e^{-\frac{\left(\frac{(2i-1)\theta}{2n} - \mu\right)^2}{2\sigma^2}}$$

$$+ \sqrt{\frac{\pi}{2}}\sigma\left(\sigma^2 + \mu^2 - 2k\theta\mu + (k\theta)^2\right)\left(1 - erf\left(\frac{\left(\frac{(2n-1)\theta}{2n} - \mu\right)}{\sqrt{2}\sigma}\right)\right) + \left(\sigma^2\left(\frac{(2n-1)\theta}{2n} + \mu - 2k\theta\right)\right) e^{-\frac{\left(\frac{(2n-1)\theta}{2n} - \mu\right)^2}{2\sigma^2}}$$

$$= \sqrt{\tfrac{\pi}{2}}\sigma\left(\sigma^2+\mu^2\right)\left(\mathrm{erf}\left(\frac{\left(\frac{\theta}{2n}-\mu\right)}{\sqrt{2}\sigma}\right)-\mathrm{erf}\left(\frac{-\mu}{\sqrt{2}\sigma}\right)\right)+\left(-\sigma^2\left(\tfrac{\theta}{2n}+\mu\right)\right)e^{-\frac{\left(\frac{\theta}{2n}-\mu\right)^2}{2\sigma^2}}+\left(\sigma^2\mu\right)e^{-\frac{\mu^2}{2\sigma^2}}$$

$$+\sum_{i=2}^{n}\sqrt{\tfrac{\pi}{2}}\sigma\left(\sigma^2+\mu^2-2k\frac{(i-1)\theta}{n}\mu+k^2\left(\frac{(i-1)\theta}{n}\right)^2\right)\mathrm{erf}\left(\frac{\left(\frac{(2i-1)\theta}{2n}-\mu\right)}{\sqrt{2}\sigma}\right)$$

$$-\sum_{i=1}^{n-1}\sqrt{\tfrac{\pi}{2}}\sigma\left(\sigma^2+\mu^2-2k\frac{i\theta}{n}\mu+k^2\left(\frac{i\theta}{n}\right)^2\right)\mathrm{erf}\left(\frac{\left(\frac{(2i-1)\theta}{2n}-\mu\right)}{\sqrt{2}\sigma}\right)$$

$$+\sum_{i=2}^{n}\left(-\sigma^2\left(\frac{(2i-1)\theta}{2n}+\mu-2k\frac{(i-1)\theta}{n}\right)\right)e^{-\frac{\left(\frac{(2i-1)\theta}{2n}-\mu\right)^2}{2\sigma^2}}+\sum_{i=1}^{n-1}\left(\sigma^2\left(\frac{(2i-1)\theta}{2n}+\mu-2k\frac{i\theta}{n}\right)\right)e^{-\frac{\left(\frac{(2i-1)\theta}{2n}-\mu\right)^2}{2\sigma^2}}$$

$$+\sqrt{\tfrac{\pi}{2}}\sigma\left(\sigma^2+\mu^2-2k\theta\mu+(k\theta)^2\right)\left(1-\mathrm{erf}\left(\frac{\left(\frac{(2n-1)\theta}{2n}-\mu\right)}{\sqrt{2}\sigma}\right)\right)+\left(\sigma^2\left(\frac{(2n-1)\theta}{2n}+\mu-2k\theta\right)\right)e^{-\frac{\left(\frac{(2n-1)\theta}{2n}-\mu\right)^2}{2\sigma^2}}$$

$$= \sqrt{\tfrac{\pi}{2}}\sigma\left(\sigma^2+\mu^2\right)\left(\mathrm{erf}\left(\frac{\left(\frac{\theta}{2n}-\mu\right)}{\sqrt{2}\sigma}\right)-\mathrm{erf}\left(\frac{-\mu}{\sqrt{2}\sigma}\right)\right)+\left(-\sigma^2\left(\tfrac{\theta}{2n}+\mu\right)\right)e^{-\frac{\left(\frac{\theta}{2n}-\mu\right)^2}{2\sigma^2}}+\left(\sigma^2\mu\right)e^{-\frac{\mu^2}{2\sigma^2}}$$

$$+\sqrt{\tfrac{\pi}{2}}\sigma\left(\sigma^2+\mu^2-2k\frac{(n-1)\theta}{n}\mu+k^2\left(\frac{(n-1)\theta}{n}\right)^2\right)\mathrm{erf}\left(\frac{\left(\frac{(2n-1)\theta}{2n}-\mu\right)}{\sqrt{2}\sigma}\right)$$

$$-\sqrt{\tfrac{\pi}{2}}\sigma\left(\sigma^2+\mu^2-2k\frac{\theta}{n}\mu+k^2\left(\frac{\theta}{n}\right)^2\right)\mathrm{erf}\left(\frac{\left(\frac{\theta}{2n}-\mu\right)}{\sqrt{2}\sigma}\right)$$

$$+\left(-\sigma^2\left(\frac{(2n-1)\theta}{2n}+\mu-2k\frac{(n-1)\theta}{n}\right)\right)e^{-\frac{\left(\frac{(2n-1)\theta}{2n}-\mu\right)^2}{2\sigma^2}}+\left(\sigma^2\left(\frac{\theta}{2n}+\mu-2k\frac{\theta}{n}\right)\right)e^{-\frac{\left(\frac{\theta}{2n}-\mu\right)^2}{2\sigma^2}}$$

$$+\sqrt{\tfrac{\pi}{2}}\sigma\left(\sigma^2+\mu^2-2k\theta\mu+(k\theta)^2\right)\left(1-\mathrm{erf}\left(\frac{\left(\frac{(2n-1)\theta}{2n}-\mu\right)}{\sqrt{2}\sigma}\right)\right)+\left(\sigma^2\left(\frac{(2n-1)\theta}{2n}+\mu-2k\theta\right)\right)e^{-\frac{\left(\frac{(2n-1)\theta}{2n}-\mu\right)^2}{2\sigma^2}}$$

$$+\sum_{i=2}^{n-1}\sqrt{\tfrac{\pi}{2}}\sigma\left(\sigma^2+\mu^2-2k\frac{(i-1)\theta}{n}\mu+k^2\left(\frac{(i-1)\theta}{n}\right)^2-\sigma^2-\mu^2+2k\frac{i\theta}{n}\mu-k^2\left(\frac{i\theta}{n}\right)^2\right)\mathrm{erf}\left(\frac{\left(\frac{(2i-1)\theta}{2n}-\mu\right)}{\sqrt{2}\sigma}\right)$$

$$+\sum_{i=2}^{n-1}\left(-\sigma^2\left(\frac{(2i-1)\theta}{2n}+\mu-2k\frac{(i-1)\theta}{n}\right)+\sigma^2\left(\frac{(2i-1)\theta}{2n}+\mu-2k\frac{i\theta}{n}\right)\right)e^{-\frac{\left(\frac{(2i-1)\theta}{2n}-\mu\right)^2}{2\sigma^2}}$$

$$= \sqrt{\tfrac{\pi}{2}}\sigma\left(\sigma^2+\mu^2\right)\left(1-\mathrm{erf}\left(\frac{-\mu}{\sqrt{2}\sigma}\right)\right)+\left(\sigma^2\mu\right)e^{-\frac{\mu^2}{2\sigma^2}}+\sqrt{\tfrac{\pi}{2}}\sigma\left(-2k\theta\mu+(k\theta)^2\right)$$

$$+\sqrt{\tfrac{\pi}{2}}\sigma\left(2k\frac{\theta}{n}\mu-k^2\theta^2\left(\frac{2n-1}{n}\right)\right)\mathrm{erf}\left(\frac{\left(\frac{(2n-1)\theta}{2n}-\mu\right)}{\sqrt{2}\sigma}\right)+\sqrt{\tfrac{\pi}{2}}\sigma\left(2k\frac{\theta}{n}\mu-k^2\left(\frac{\theta}{n}\right)^2\right)\mathrm{erf}\left(\frac{\left(\frac{\theta}{2n}-\mu\right)}{\sqrt{2}\sigma}\right)$$

$$+\left(-\sigma^2\left(2k\frac{\theta}{n}\right)\right)e^{-\frac{\left(\frac{(2n-1)\theta}{2n}-\mu\right)^2}{2\sigma^2}}+\left(\sigma^2\left(-2k\frac{\theta}{n}\right)\right)e^{-\frac{\left(\frac{\theta}{2n}-\mu\right)^2}{2\sigma^2}}$$

$$+\sum_{i=2}^{n-1}\sqrt{\tfrac{\pi}{2}}\sigma\left(2k\frac{\theta}{n}\mu-k^2\left(\frac{\theta}{n}\right)^2(2i-1)\right)\mathrm{erf}\left(\frac{\left(\frac{(2i-1)\theta}{2n}-\mu\right)}{\sqrt{2}\sigma}\right)$$

$$+\sum_{i=2}^{n-1}\left(-\sigma^2 2k\frac{\theta}{n}\right)e^{-\frac{\left(\frac{(2i-1)\theta}{2n}-\mu\right)^2}{2\sigma^2}}$$

$$= \sqrt{\tfrac{\pi}{2}}\sigma \left(\sigma^2 + \mu^2\right)\left(1 - \mathrm{erf}\left(\frac{-\mu}{\sqrt{2}\sigma}\right)\right) + \left(\sigma^2\mu\right)e^{-\frac{\mu^2}{2\sigma^2}} + \sqrt{\tfrac{\pi}{2}}\sigma\left(-2k\theta\mu + (k\theta)^2\right)$$

$$+ \sum_{i=1}^{n}\sqrt{\tfrac{\pi}{2}}\sigma\left(2k\frac{\theta}{n}\mu - k^2\left(\frac{\theta}{n}\right)^2(2i-1)\right)\mathrm{erf}\left(\frac{\left(\frac{(2i-1)\theta}{2n} - \mu\right)}{\sqrt{2}\sigma}\right)$$

$$+ \sum_{i=1}^{n}\left(-\sigma^2 2k\frac{\theta}{n}\right)e^{-\frac{\left(\frac{(2i-1)\theta}{2n} - \mu\right)^2}{2\sigma^2}}$$

$$= k^2\left(\sqrt{\tfrac{\pi}{2}}\sigma\theta^2 - \sqrt{\tfrac{\pi}{2}}\sigma\left(\frac{\theta}{n}\right)^2\sum_{i=1}^{n}(2i-1)\mathrm{erf}\left(\frac{\left(\frac{(2i-1)\theta}{2n} - \mu\right)}{\sqrt{2}\sigma}\right)\right)$$

$$+ k\left(2\sqrt{\tfrac{\pi}{2}}\sigma\frac{\theta}{n}\mu\sum_{i=1}^{n}\mathrm{erf}\left(\frac{\left(\frac{(2i-1)\theta}{2n} - \mu\right)}{\sqrt{2}\sigma}\right) - 2\sqrt{\tfrac{\pi}{2}}\sigma\theta\mu - 2\sigma^2\theta\sum_{i=1}^{n}\frac{1}{n}e^{-\frac{\left(\frac{(2i-1)\theta}{2n} - \mu\right)^2}{2\sigma^2}}\right)$$

$$+ \sqrt{\tfrac{\pi}{2}}\sigma\left(\sigma^2 + \mu^2\right)\left(1 - \mathrm{erf}\left(\frac{-\mu}{\sqrt{2}\sigma}\right)\right) + \left(\sigma^2\mu\right)e^{-\frac{\mu^2}{2\sigma^2}}$$

Since this is a quadratic function with respect to $k$, the value of $k$ that minimizes it is:

$$k = -\frac{2\sqrt{\tfrac{\pi}{2}}\sigma\theta\mu\left(\sum_{i=1}^{n}\frac{1}{n}\mathrm{erf}\left(\frac{\left(\frac{(2i-1)\theta}{2n} - \mu\right)}{\sqrt{2}\sigma}\right) - 1\right) - 2\sigma^2\theta\sum_{i=1}^{n}\frac{1}{n}e^{-\frac{\left(\frac{(2i-1)\theta}{2n} - \mu\right)^2}{2\sigma^2}}}{2\sqrt{\tfrac{\pi}{2}}\sigma\theta^2\left(1 - \sum_{i=1}^{n}\left(\frac{2i-1}{n^2}\right)\mathrm{erf}\left(\frac{\left(\frac{(2i-1)\theta}{2n} - \mu\right)}{\sqrt{2}\sigma}\right)\right)}$$

$$= \frac{\mu\left(1 - \sum_{i=1}^{n}\frac{1}{n}\mathrm{erf}\left(\frac{\left(\frac{(2i-1)\theta}{2n} - \mu\right)}{\sqrt{2}\sigma}\right)\right)}{\theta\left(1 - \sum_{i=0}^{n}\left(\frac{2i-1}{n^2}\right)\mathrm{erf}\left(\frac{\left(\frac{(2i-1)\theta}{2n} - \mu\right)}{\sqrt{2}\sigma}\right)\right)} + \frac{\sigma\sum_{i=1}^{n}\frac{1}{n}e^{-\frac{\left(\frac{(2i-1)\theta}{2n} - \mu\right)^2}{2\sigma^2}}}{\sqrt{\tfrac{\pi}{2}}\theta\left(1 - \sum_{i=1}^{n}\left(\frac{2i-1}{n^2}\right)\mathrm{erf}\left(\frac{\left(\frac{(2i-1)\theta}{2n} - \mu\right)}{\sqrt{2}\sigma}\right)\right)}$$

$$\tag{78}$$

$\square$

## H. Proof of Lemma 4.9

**Lemma H.1.** *(Repeated from Lemma 4.9)*

$$QE_2(\theta, k) = \int_{-\infty}^{+\infty}\left(f_2(x, \theta, k) - \max(x, 0)\right)^2 e^{-\frac{(x-\mu)^2}{2\sigma^2}}\,dx \tag{79}$$

$$f_2(x, \theta, k) = \frac{\theta}{N}clamp\left(\lfloor\frac{Nx + \frac{k\theta}{2}}{k\theta}\rfloor, 0, N\right) \tag{80}$$

*When $\theta$ is fixed, $QE_2(\theta, k)$ reaches its minimum value when $k = 1$.*

*Proof.* Expand the calculation of $QE_2$:

$$\int_{-\infty}^{+\infty}\left(\frac{\theta}{n}clamp\left(\lfloor\frac{nx + \frac{k\theta}{2}}{k\theta}\rfloor, 0, n\right) - \max(x, 0)\right)^2 e^{-\frac{(x-\mu)^2}{2\sigma^2}}\,dx$$

$$= \int_{0}^{\frac{\theta k}{2n}}(x)^2 e^{-\frac{(x-\mu)^2}{2\sigma^2}}\,dx + \sum_{i=1}^{n-1}\int_{\frac{(2i-1)\theta k}{2n}}^{\frac{(2i+1)\theta k}{2n}}\left(\frac{i\theta}{n} - x\right)^2 e^{-\frac{(x-\mu)^2}{2\sigma^2}}\,dx + \int_{\frac{(2n-1)\theta k}{2n}}^{+\infty}(\theta - x)^2 e^{-\frac{(x-\mu)^2}{2\sigma^2}}\,dx$$

$$\tag{81}$$

$$
= \sqrt{\tfrac{\pi}{2}}\sigma\left(\sigma^2+\mu^2\right)\left(\mathrm{erf}\left(\frac{\left(\frac{\theta k}{2n}-\mu\right)}{\sqrt{2}\sigma}\right)-\mathrm{erf}\left(\frac{-\mu}{\sqrt{2}\sigma}\right)\right)+\left(-\sigma^2\left(\tfrac{\theta k}{2n}+\mu\right)\right)e^{-\frac{\left(\frac{\theta k}{2n}-\mu\right)^2}{2\sigma^2}}+\left(\sigma^2\mu\right)e^{-\frac{\mu^2}{2\sigma^2}}
$$

$$
+\sum_{i=1}^{n-1}\sqrt{\tfrac{\pi}{2}}\sigma\left(\sigma^2+\mu^2-2\frac{i\theta}{n}\mu+\left(\frac{i\theta}{n}\right)^2\right)\left(\mathrm{erf}\left(\frac{\left(\frac{(2i+1)\theta k}{2n}-\mu\right)}{\sqrt{2}\sigma}\right)-\mathrm{erf}\left(\frac{\left(\frac{(2i-1)\theta k}{2n}-\mu\right)}{\sqrt{2}\sigma}\right)\right)
$$

$$
+\sum_{i=1}^{n-1}\left(-\sigma^2\left(\tfrac{(2i+1)\theta k}{2n}+\mu-2\frac{i\theta}{n}\right)\right)e^{-\frac{\left(\frac{(2i+1)\theta k}{2n}-\mu\right)^2}{2\sigma^2}}+\sum_{i=1}^{n-1}\left(\sigma^2\left(\tfrac{(2i-1)\theta k}{2n}+\mu-2\frac{i\theta}{n}\right)\right)e^{-\frac{\left(\frac{(2i-1)\theta k}{2n}-\mu\right)^2}{2\sigma^2}}
$$

$$
+\sqrt{\tfrac{\pi}{2}}\sigma\left(\sigma^2+\mu^2-2\theta\mu+\theta^2\right)\left(1-\mathrm{erf}\left(\frac{\left(\frac{(2n-1)\theta k}{2n}-\mu\right)}{\sqrt{2}\sigma}\right)\right)+\left(\sigma^2\left(\tfrac{(2n-1)\theta k}{2n}+\mu-2\theta\right)\right)e^{-\frac{\left(\frac{(2n-1)\theta k}{2n}-\mu\right)^2}{2\sigma^2}}
$$

$$
= \sqrt{\tfrac{\pi}{2}}\sigma\left(\sigma^2+\mu^2\right)\left(\mathrm{erf}\left(\frac{\left(\frac{\theta k}{2n}-\mu\right)}{\sqrt{2}\sigma}\right)-\mathrm{erf}\left(\frac{-\mu}{\sqrt{2}\sigma}\right)\right)+\left(-\sigma^2\left(\tfrac{\theta k}{2n}+\mu\right)\right)e^{-\frac{\left(\frac{\theta k}{2n}-\mu\right)^2}{2\sigma^2}}+\left(\sigma^2\mu\right)e^{-\frac{\mu^2}{2\sigma^2}}
$$

$$
+\sqrt{\tfrac{\pi}{2}}\sigma\left(\sigma^2+\mu^2-2\theta\mu+\theta^2\right)\left(1-\mathrm{erf}\left(\frac{\left(\frac{(2n-1)\theta k}{2n}-\mu\right)}{\sqrt{2}\sigma}\right)\right)+\left(\sigma^2\left(\tfrac{(2n-1)\theta k}{2n}+\mu-2\theta\right)\right)\theta e^{-\frac{\left(\frac{(2n-1)\theta k}{2n}-\mu\right)^2}{2\sigma^2}}
$$

$$
+\sum_{i=2}^{n}\sqrt{\tfrac{\pi}{2}}\sigma\left(\sigma^2+\mu^2-2\frac{(i-1)\theta}{n}\mu+\left(\frac{(i-1)\theta}{n}\right)^2\right)\mathrm{erf}\left(\frac{\left(\frac{(2i-1)\theta k}{2n}-\mu\right)}{\sqrt{2}\sigma}\right)
$$

$$
-\sum_{i=1}^{n-1}\sqrt{\tfrac{\pi}{2}}\sigma\left(\sigma^2+\mu^2-2\frac{i\theta}{n}\mu+\left(\frac{i\theta}{n}\right)^2\right)\mathrm{erf}\left(\frac{\left(\frac{(2i-1)\theta k}{2n}-\mu\right)}{\sqrt{2}\sigma}\right)
$$

$$
+\sum_{i=2}^{n}\left(-\sigma^2\left(\tfrac{(2i-1)\theta k}{2n}+\mu-2\frac{(i-1)\theta}{n}\right)\right)e^{-\frac{\left(\frac{(2i-1)\theta k}{2n}-\mu\right)^2}{2\sigma^2}}+\sum_{i=1}^{n-1}\left(\sigma^2\left(\tfrac{(2i-1)\theta k}{2n}+\mu-2\frac{i\theta}{n}\right)\right)e^{-\frac{\left(\frac{(2i-1)\theta k}{2n}-\mu\right)^2}{2\sigma^2}}
$$

$$
= \sqrt{\tfrac{\pi}{2}}\sigma\left(\sigma^2+\mu^2\right)\left(\mathrm{erf}\left(\frac{\left(\frac{\theta k}{2n}-\mu\right)}{\sqrt{2}\sigma}\right)-\mathrm{erf}\left(\frac{-\mu}{\sqrt{2}\sigma}\right)\right)+\left(-\sigma^2\left(\tfrac{\theta k}{2n}+\mu\right)\right)e^{-\frac{\left(\frac{\theta k}{2n}-\mu\right)^2}{2\sigma^2}}+\left(\sigma^2\mu\right)e^{-\frac{\mu^2}{2\sigma^2}}
$$

$$
+\sqrt{\tfrac{\pi}{2}}\sigma\left(\sigma^2+\mu^2-2\theta\mu+\theta^2\right)\left(1-\mathrm{erf}\left(\frac{\left(\frac{(2n-1)\theta k}{2n}-\mu\right)}{\sqrt{2}\sigma}\right)\right)+\left(\sigma^2\left(\tfrac{(2n-1)\theta k}{2n}+\mu-2\theta\right)\right)\theta e^{-\frac{\left(\frac{(2n-1)\theta k}{2n}-\mu\right)^2}{2\sigma^2}}
$$

$$
+\sqrt{\tfrac{\pi}{2}}\sigma\left(\sigma^2+\mu^2-2\frac{(n-1)\theta}{n}\mu+\left(\frac{(n-1)\theta}{n}\right)^2\right)\mathrm{erf}\left(\frac{\left(\frac{(2n-1)\theta k}{2n}-\mu\right)}{\sqrt{2}\sigma}\right)-\sqrt{\tfrac{\pi}{2}}\sigma\left(\sigma^2+\mu^2-2\frac{\theta}{n}\mu+\left(\frac{\theta}{n}\right)^2\right)\mathrm{erf}\left(\frac{\left(\frac{\theta k}{2n}-\mu\right)}{\sqrt{2}\sigma}\right)
$$

$$
+\sum_{i=2}^{n-1}\sqrt{\tfrac{\pi}{2}}\sigma\left(-2\frac{(i-1)\theta}{n}\mu+\left(\frac{(i-1)\theta}{n}\right)^2+2\frac{i\theta}{n}\mu-\left(\frac{i\theta}{n}\right)^2\right)\mathrm{erf}\left(\frac{\left(\frac{(2i-1)\theta k}{2n}-\mu\right)}{\sqrt{2}\sigma}\right)
$$

$$
+\left(-\sigma^2\left(\tfrac{(2n-1)\theta k}{2n}+\mu-2\frac{(n-1)\theta}{n}\right)\right)e^{-\frac{\left(\frac{(2n-1)\theta k}{2n}-\mu\right)^2}{2\sigma^2}}+\left(\sigma^2\left(\tfrac{\theta k}{2n}+\mu-2\frac{\theta}{n}\right)\right)e^{-\frac{\left(\frac{\theta k}{2n}-\mu\right)^2}{2\sigma^2}}
$$

$$
+\sum_{i=2}^{n-1}\left(-\sigma^2\left(\tfrac{(2i-1)\theta k}{2n}+\mu-2\frac{(i-1)\theta}{n}\right)+\sigma^2\left(\tfrac{(2i-1)\theta k}{2n}+\mu-2\frac{i\theta}{n}\right)\right)e^{-\frac{\left(\frac{(2i-1)\theta k}{2n}-\mu\right)^2}{2\sigma^2}}
$$

$$
= \sqrt{\tfrac{\pi}{2}}\sigma\left(\sigma^2+\mu^2\right)\left(-\mathrm{erf}\left(\frac{-\mu}{\sqrt{2}\sigma}\right)\right)+\left(\sigma^2\mu\right)e^{-\frac{\mu^2}{2\sigma^2}}+\sqrt{\tfrac{\pi}{2}}\sigma\left(\sigma^2+\mu^2-2\theta\mu+\theta^2\right)
$$

$$
+\sqrt{\tfrac{\pi}{2}}\sigma\left(2\frac{\theta}{n}\mu+\left(\frac{\theta}{n}\right)^2(-2i+1)\right)\mathrm{erf}\left(\frac{\left(\frac{(2n-1)\theta k}{2n}-\mu\right)}{\sqrt{2}\sigma}\right)+\sqrt{\tfrac{\pi}{2}}\sigma\left(2\frac{\theta}{n}\mu-\left(\frac{\theta}{n}\right)^2\right)\mathrm{erf}\left(\frac{\left(\frac{\theta k}{2n}-\mu\right)}{\sqrt{2}\sigma}\right)
$$

$$
+\sum_{i=2}^{n-1}\sqrt{\tfrac{\pi}{2}}\sigma\left(2\frac{\theta}{n}\mu+\left(\frac{\theta}{n}\right)^2(-2i+1)\right)\mathrm{erf}\left(\frac{\left(\frac{(2i-1)\theta k}{2n}-\mu\right)}{\sqrt{2}\sigma}\right)
$$

$$
+\left(-2\sigma^2\frac{\theta}{n}\right)e^{-\frac{\left(\frac{(2n-1)\theta k}{2n}-\mu\right)^2}{2\sigma^2}}+\left(\sigma^2\left(-2\frac{\theta}{n}\right)\right)e^{-\frac{\left(\frac{\theta k}{2n}-\mu\right)^2}{2\sigma^2}}
$$

$$
+\sum_{i=2}^{n-1}\left(-2\sigma^2\frac{\theta}{n}\right)e^{-\frac{\left(\frac{(2i-1)\theta k}{2n}-\mu\right)^2}{2\sigma^2}}
$$

$$= \sqrt{\tfrac{\pi}{2}}\sigma\left(\sigma^2 + \mu^2\right)\left(1 - \mathrm{erf}\left(\frac{-\mu}{\sqrt{2}\sigma}\right)\right) + \left(\sigma^2\mu\right)e^{-\frac{\mu^2}{2\sigma^2}} + \sqrt{\tfrac{\pi}{2}}\sigma\left(-2\theta\mu + \theta^2\right)$$

$$+ \sum_{i=1}^{n}\sqrt{\tfrac{\pi}{2}}\sigma\left(2\frac{\theta}{n}\mu + \left(\frac{\theta}{n}\right)^2(-2i+1)\right)\mathrm{erf}\left(\frac{\left(\frac{(2i-1)\theta k}{2n} - \mu\right)}{\sqrt{2}\sigma}\right) + \sum_{i=1}^{n}\left(-2\sigma^2\frac{\theta}{n}\right)e^{-\frac{\left(\frac{(2i-1)\theta k}{2n} - \mu\right)^2}{2\sigma^2}}$$

$$= \sqrt{\tfrac{\pi}{2}}\sigma\left(\sigma^2 + \mu^2\right)\left(1 - \mathrm{erf}\left(\frac{-\mu}{\sqrt{2}\sigma}\right)\right) + \left(\sigma^2\mu\right)e^{-\frac{\mu^2}{2\sigma^2}} + \sqrt{\tfrac{\pi}{2}}\sigma\left(-2\theta\mu + \theta^2\right)$$

$$+ 2\sigma\frac{\theta}{n}\sum_{i=1}^{n}\left(\sqrt{\tfrac{\pi}{2}}\mu\,\mathrm{erf}\left(\frac{\left(\frac{(2i-1)\theta k}{2n} - \mu\right)}{\sqrt{2}\sigma}\right) + \sqrt{\tfrac{\pi}{2}}\theta\left(\frac{2i-1}{n}\right)\mathrm{erf}\left(\frac{\left(\frac{(2i-1)\theta k}{2n} - \mu\right)}{\sqrt{2}\sigma}\right) - \sigma e^{-\frac{\left(\frac{(2i-1)\theta k}{2n} - \mu\right)^2}{2\sigma^2}}\right)$$

We only need to extract the part containing $k$, denoted as $L$, for calculation.

$$L = \sum_{i=1}^{n}\left(\sqrt{\tfrac{\pi}{2}}\mu\,\mathrm{erf}\left(\frac{\left(\frac{(2i-1)\theta k}{2n} - \mu\right)}{\sqrt{2}\sigma}\right) + \sqrt{\tfrac{\pi}{2}}\left(-\frac{(2i-1)\,\theta}{2n}\right)\mathrm{erf}\left(\frac{\left(\frac{(2i-1)\theta k}{2n} - \mu\right)}{\sqrt{2}\sigma}\right) - \sigma e^{-\frac{\left(\frac{(2i-1)\theta k}{2n} - \mu\right)^2}{2\sigma^2}}\right) \tag{82}$$

Since $\mathrm{erf}'(x) = \frac{2}{\sqrt{\pi}}e^{-x^2}$,

$$L' = \left(\sum_{i=1}^{n}\sqrt{2}\mu e^{-\frac{\left(\frac{(2i-1)\theta k}{2n} - \mu\right)^2}{2\sigma^2}}\left(\frac{\frac{(2i-1)\theta}{2n}}{\sqrt{2}\sigma}\right) + \sqrt{\tfrac{\pi}{2}}\left(-\frac{(2i-1)\,\theta}{2n}\right)\frac{2}{\sqrt{\pi}}e^{-\frac{\left(\frac{(2i-1)\theta k}{2n} - \mu\right)^2}{2\sigma^2}}\left(\frac{\frac{(2i-1)\theta}{2n}}{\sqrt{2}\sigma}\right) - \sigma e^{-\frac{\left(\frac{(2i-1)\theta k}{2n} - \mu\right)^2}{2\sigma^2}}\left(-\frac{2\left(\frac{(2i-1)\theta k}{2n} - \mu\right)}{2\sigma^2}\right)\frac{(2i-1)\theta}{2n}\right)$$

$$= \sum_{i=1}^{n}\mu\frac{(2i-1)\theta}{2n\sigma}e^{-\frac{\left(\frac{(2i-1)\theta k}{2n} - \mu\right)^2}{2\sigma^2}} - \frac{(2i-1)\theta}{2n\sigma}\left(\frac{(2i-1)\,\theta}{2n}\right)e^{-\frac{\left(\frac{(2i-1)\theta k}{2n} - \mu\right)^2}{2\sigma^2}} + \left(\frac{(2i-1)\theta k}{2n} - \mu\right)\frac{(2i-1)\theta}{2n\sigma}e^{-\frac{\left(\frac{(2i-1)\theta k}{2n} - \mu\right)^2}{2\sigma^2}}$$

$$= \sum_{i=1}^{n} -\frac{(2i-1)\theta}{2n\sigma}\frac{(2i-1)\,\theta}{2n}e^{-\frac{\left(\frac{(2i-1)\theta k}{2n} - \mu\right)^2}{2\sigma^2}} + \frac{(2i-1)\theta k}{2n}\frac{(2i-1)\theta}{2n\sigma}e^{-\frac{\left(\frac{(2i-1)\theta k}{2n} - \mu\right)^2}{2\sigma^2}}$$

$$= \sum_{i=1}^{n}\frac{(2i-1)^2\,\theta^2}{4n^2\sigma}(k-1)\,e^{-\frac{\left(\frac{(2i-1)\theta k}{2n} - \mu\right)^2}{2\sigma^2}} \tag{83}$$

So, the value of $k$ that minimizes the function is $k = 1$. $\qquad\square$

## I. Proof of Theorem 4.10

**Theorem I.1.** (*Repeated from Theorem 4.10*) *Starting from any positive initial value of $\theta$, the rate of change $k_1$ can be continuously calculated based on the prior mean $\mu$, variance $\sigma^2$, and the current threshold $\theta$ using Equation (27). The iteration $\theta = k_1\theta$ continues until convergence, at which point the global optimal threshold $\theta$ is obtained. The process is guaranteed to converge as long as the threshold is greater than 0.*

*Proof.* Assume the optimal threshold is $\theta$, then according to Lemma 4.8, after one update $k\theta$ should be equal to $\theta$, that is $k = 1$.

To prove that $\theta$ is the optimal value, it is equivalent to proving that the following equation has an unique solution:

$$\frac{\mu}{\theta}\frac{1 - \sum_{i=1}^{n}\frac{1}{n}\mathrm{erf}\left(\frac{\left(\frac{(2i-1)\theta}{2n} - \mu\right)}{\sqrt{2}\sigma}\right)}{1 - \sum_{i=1}^{n}\frac{2i-1}{n^2}\mathrm{erf}\left(\frac{\left(\frac{(2i-1)\theta}{2n} - \mu\right)}{\sqrt{2}\sigma}\right)} + \frac{\sigma}{\sqrt{\tfrac{\pi}{2}}\theta}\frac{\sum_{i=1}^{n}\frac{1}{n}e^{-\frac{\left(\frac{(2i-1)\theta}{2n} - \mu\right)^2}{2\sigma^2}}}{1 - \sum_{i=1}^{n}\frac{2i-1}{n^2}\mathrm{erf}\left(\frac{\left(\frac{(2i-1)\theta}{2n} - \mu\right)}{\sqrt{2}\sigma}\right)} = 1 \tag{84}$$

It is equivalent to proving that $f(\theta)$ has an unique root.

$$f(\theta) = \mu\left(1 - \sum_{i=1}^{n}\frac{1}{n}\mathrm{erf}\left(\frac{\left(\frac{(2i-1)\theta}{2n}-\mu\right)}{\sqrt{2}\sigma}\right)\right) + \frac{\sigma}{\sqrt{\frac{\pi}{2}}}\sum_{i=1}^{n}\frac{1}{n}e^{-\frac{\left(\frac{(2i-1)\theta}{2n}-\mu\right)^2}{2\sigma^2}} - \theta\left(1 - \sum_{i=1}^{n}\frac{2i-1}{n^2}\mathrm{erf}\left(\frac{\left(\frac{(2i-1)\theta}{2n}-\mu\right)}{\sqrt{2}\sigma}\right)\right) = 0 \quad (85)$$

**Step 1: Calculate the first derivative of $f$**

Since $\mathrm{erf}'(x) = \frac{2}{\sqrt{\pi}}e^{-x^2}$,

$$
\begin{aligned}
f'(\theta) &= -\frac{2\mu}{\sqrt{\pi}}\sum_{i=1}^{n}\frac{1}{n}e^{-\frac{\left(\frac{(2i-1)\theta}{2n}-\mu\right)^2}{2\sigma^2}}\left(\frac{(2i-1)}{2n\sqrt{2}\sigma}\right) + \frac{\sigma}{\sqrt{\frac{\pi}{2}}}\sum_{i=1}^{n}\frac{1}{n}e^{-\frac{\left(\frac{(2i-1)\theta}{2n}-\mu\right)^2}{2\sigma^2}}\left(\frac{-2\left(\frac{(2i-1)\theta}{2n}-\mu\right)}{2\sigma^2}\right)\left(\frac{(2i-1)}{2n}\right) \\
&\quad -1 + \sum_{i=1}^{n}\frac{2i-1}{n^2}\mathrm{erf}\left(\frac{\left(\frac{(2i-1)\theta}{2n}-\mu\right)}{\sqrt{2}\sigma}\right) + \theta\sum_{i=1}^{n}\frac{2i-1}{n^2}\frac{2}{\sqrt{\pi}}e^{-\frac{\left(\frac{(2i-1)\theta}{2n}-\mu\right)^2}{2\sigma^2}}\left(\frac{(2i-1)}{2n\sqrt{2}\sigma}\right) \\
&= \frac{2}{\sqrt{\pi}}\sum_{i=1}^{n}\left(-\mu\frac{1}{n}+\theta\frac{2i-1}{n^2}-\frac{\left(\frac{(2i-1)\theta}{2n}-\mu\right)}{n}\right)\left(\frac{(2i-1)}{2n\sqrt{2}\sigma}\right)e^{-\frac{\left(\frac{(2i-1)\theta}{2n}-\mu\right)^2}{2\sigma^2}} - 1 + \sum_{i=1}^{n}\frac{2i-1}{n^2}\mathrm{erf}\left(\frac{\left(\frac{(2i-1)\theta}{2n}-\mu\right)}{\sqrt{2}\sigma}\right) \\
&= -1 + \sum_{i=1}^{n}\frac{2i-1}{n^2}\mathrm{erf}\left(\frac{\left(\frac{(2i-1)\theta}{2n}-\mu\right)}{\sqrt{2}\sigma}\right) + \frac{2\theta}{\sqrt{\pi}}\sum_{i=1}^{n}\left(\frac{2i-1}{2n^2}\right)\left(\frac{(2i-1)}{2n\sqrt{2}\sigma}\right)e^{-\frac{\left(\frac{(2i-1)\theta}{2n}-\mu\right)^2}{2\sigma^2}}
\end{aligned}
\quad (86)
$$

**Step 2: Calculate the second derivative of $f$**

$$
\begin{aligned}
f''(\theta) &= \frac{2}{\sqrt{\pi}}\sum_{i=1}^{n}\frac{2i-1}{n^2}e^{-\left(\frac{\left(\frac{(2i-1)\theta}{2n}-\mu\right)}{\sqrt{2}\sigma}\right)^2}\left(\frac{(2i-1)}{2n\sqrt{2}\sigma}\right) + \frac{2}{\sqrt{\pi}}\sum_{i=1}^{n}\left(\frac{2i-1}{2n^2}\right)\left(\frac{(2i-1)}{2n\sqrt{2}\sigma}\right)e^{-\frac{\left(\frac{(2i-1)\theta}{2n}-\mu\right)^2}{2\sigma^2}} \\
&\quad + \frac{\theta}{\sqrt{\pi}}\sum_{i=1}^{n}\left(\frac{2i-1}{n^2}\right)\left(\frac{(2i-1)}{2n\sqrt{2}\sigma}\right)e^{-\frac{\left(\frac{(2i-1)\theta}{2n}-\mu\right)^2}{2\sigma^2}}\left(\frac{-2\left(\frac{(2i-1)\theta}{2n}-\mu\right)}{2\sigma^2}\right)\left(\frac{(2i-1)}{2n}\right) \\
&= \frac{1}{\sqrt{\pi}}\sum_{i=1}^{n}\frac{2i-1}{n^2}e^{-\left(\frac{\left(\frac{(2i-1)\theta}{2n}-\mu\right)}{\sqrt{2}\sigma}\right)^2}\left(\frac{(2i-1)}{2n\sqrt{2}\sigma}\right)\left(3 - \theta\left(\frac{\left(\frac{(2i-1)\theta}{2n}-\mu\right)}{\sigma^2}\right)\left(\frac{(2i-1)}{2n}\right)\right)
\end{aligned}
\quad (87)
$$

**Step 3: Analyze the trend of the second derivative $f''(\theta)$**

To analyze the changing trend of $f''(x)$ on the interval from 0 to positive infinity, we start by examining the given expressions:

$$f''(0) = \frac{3}{\sqrt{\pi}}\sum_{i=1}^{n}\frac{2i-1}{n^2}e^{-\left(\frac{\mu}{\sqrt{2}\sigma}\right)^2}\left(\frac{2i-1}{2n\sqrt{2}\sigma}\right) > 0 \quad (88)$$

and

$$f''(+\infty) = 0^- \quad (89)$$

The behavior of $f''(\theta)$ as $\theta$ increases from 0 to $+\infty$ depends on two parts: $e^{-\left(\frac{\left(\frac{(2i-1)\theta}{2n}-\mu\right)}{\sqrt{2}\sigma}\right)^2}$ and $3 - \theta\left(\frac{\left(\frac{(2i-1)\theta}{2n}-\mu\right)}{\sigma^2}\right)\left(\frac{2i-1}{2n}\right)$. Since $e^{-\left(\frac{\left(\frac{(2i-1)\theta}{2n}-\mu\right)}{\sqrt{2}\sigma}\right)^2} > 0$, the sign of $f''(\theta)$ is determined by the second part. This term decreases from 3 to negative infinity as $\theta$ increases.

Thus, $f''(\theta)$ initially decreases from a positive number to a negative number and then continues to increase within the negative range.

Step 4: Analyze the Trend of the First Derivative $f'(\theta)$

Given:

$$f'(0) = -1 + \mathrm{erf}\left(\frac{-\mu}{\sqrt{2}\sigma}\right) < 0, \ f'(+\infty) = 0 \tag{90}$$

Since $f''(\theta)$ first decreases from a positive number to a negative number and then increases within the negative range, $f'(\theta)$ will first increase from a negative number to a positive number and then decrease back to zero.

Step 5: Analyze the Trend of the Function $f(\theta)$

Given:

$$f(0) = \mu\left(1 - \sum_{i=1}^{n}\frac{1}{n}\mathrm{erf}\left(\frac{-\mu}{\sqrt{2}\sigma}\right)\right) + \frac{\sigma}{\sqrt{\frac{\pi}{2}}}\sum_{i=1}^{n}\frac{1}{n}e^{-\frac{\mu^2}{2\sigma^2}} = \mu\left(1 - \mathrm{erf}\left(\frac{-\mu}{\sqrt{2}\sigma}\right)\right) + \frac{\sigma}{\sqrt{\frac{\pi}{2}}}e^{-\frac{\mu^2}{2\sigma^2}} > 0 \tag{91}$$

$$f(+\infty) = \mu(1-1) + \frac{\sigma}{\sqrt{\frac{\pi}{2}}}\sum_{i=1}^{n}\frac{1}{n}\cdot 0 - \infty\left(1 - \sum_{i=1}^{n}\frac{2i-1}{n^2}\mathrm{erf}\left(\frac{\left(\frac{(2i-1)\infty}{2n}\right) - \mu}{\sqrt{2}\sigma}\right)\right) = 0 + 0 - 0 = 0 \tag{92}$$

Since $f'(\theta)$ first increases from a negative number to a positive number and then decreases back to zero, $f(\theta)$ will first decrease from a positive number to a minimum value and then increase towards zero.

Therefore, $f(\theta)$ has a unique root, which implies that the local optimal threshold $\theta$ is the global optimal threshold. $\qquad\square$

## J. The Employed Neuron Model

We use a differential version of the Multi-Threshold (MT) neuron, as introduced in (Huang et al., 2024). The differential MT neuron is characterized by several parameters, including the base threshold $\theta$, and a total of $2n$ thresholds, with $n$ positive and $n$ negative thresholds. The threshold values of the differential MT neuron are indexed by $i$, where $\lambda_i^l$ represents the $i$-th threshold value in the layer $l$:

$$\lambda_1^l = \theta^l, \lambda_2^l = \frac{\theta^l}{2}, ..., \lambda_n^l = \frac{\theta^l}{2^{n-1}},$$
$$\lambda_{n+1}^l = -\theta^l, \lambda_{n+2}^l = -\frac{\theta^l}{2}, ..., \lambda_{2n}^l = -\frac{\theta^l}{2^{n-1}}. \tag{93}$$

Let variables $\boldsymbol{I}^l[t]$, $\boldsymbol{s}_i^l[t]$, $\boldsymbol{x}^l[t]$, $\boldsymbol{m}^l[t]$, $\boldsymbol{v}^l[t]$, and $\boldsymbol{m}_r^l[t]$ represent the input current, output spike of the $i-th$ threshold, encoded output value, the membrane potential before and after spikes in the $l$-th layer at time-step $t$, and another membrane potential to record encoded input rate information, respectively. The dynamics of the MT neurons are described by the following equations:

$$\boldsymbol{I}^l[t] = \boldsymbol{m}_r^l[t] + \boldsymbol{x}^{l-1}[t] \tag{94}$$

$$\boldsymbol{m}_r^l[t+1] = \boldsymbol{m}_r^l[t] + \frac{\boldsymbol{x}^{l-1}[t]}{t} - \frac{\boldsymbol{x}^l[t]}{t} \tag{95}$$

$$\boldsymbol{m}^l[t] = \boldsymbol{v}^l[t-1] + \boldsymbol{I}^l[t], \tag{96}$$

$$\boldsymbol{s}_i^l[t] = \mathrm{MTH}_{\theta,n}(\boldsymbol{m}^l[t], i) \tag{97}$$

$$\boldsymbol{x}^l[t] = \sum_i \boldsymbol{s}_i^l[t]\lambda_i^l, \tag{98}$$

$$\boldsymbol{v}^l[t] = \boldsymbol{m}^l[t] - \boldsymbol{x}^l[t], \tag{99}$$

$$\mathrm{MTH}_{\theta,n}(\boldsymbol{m}^l[t], i) = \begin{cases} 0, \text{if} & \lambda_{2n} < \boldsymbol{x} < \lambda_n \\ 1, \text{elif} & i = \arg\min_p |\boldsymbol{x} - \lambda_p| \\ 0, \text{else} \end{cases} . \tag{100}$$

When $n = 1$, this model reduces to a differential IF neuron with a negative threshold.

## K. Result of Different Models on ImageNet Dataset

Table 3 and 4 present the evaluation results for various CNN-based and Transformer-based models. The variable $2n$ denotes the number of positive and negative thresholds in the multi-threshold neurons, where the negative thresholds are the opposite number of corresponding positive thresholds. The energy ratio is the energy consumption of SNNs divided by that of ANNs. For the ResNet18, ResNet34, and VGG16 models, the threshold scale is set to 1. For the ViT and EVA02 models, the threshold scale is 4.

*Table 3.* Accuracy and Energy Efficiency of DCGS(Ours) on CNN-based models for ImageNet Dataset

| Architecture/Parameter(M) | Original(ANN)(%) | n | Accuracy / Energy | Time-step $T$ | | | | | |
|---|---|---|---|---|---|---|---|---|---|
| | | | | 2 | 4 | 8 | 16 | 32 | 64 |
| ResNet18 / 11.7M | 71.49 | 1 | Acc | 0.10 | 0.11 | 1.57 | 51.89 | 69.89 | 71.08 |
| | | | Energy ratio | 0.05 | 0.09 | 0.18 | 0.31 | 0.49 | 0.76 |
| | | 4 | Acc | 61.45 | 70.07 | 71.31 | 71.47 | 71.49 | - |
| | | | Energy ratio | 0.14 | 0.22 | 0.33 | 0.46 | 0.66 | - |
| | | 8 | Acc | 65.30 | 70.96 | 71.40 | 71.51 | - | - |
| | | | Energy ratio | 0.17 | 0.32 | 0.48 | 0.63 | - | - |
| ResNet34 / 21.8M | 76.42 | 1 | Acc | 0.10 | 0.14 | 0.46 | 8.76 | 58.86 | 74.11 |
| | | | Energy ratio | 0.04 | 0.09 | 0.19 | 0.34 | 0.61 | 0.97 |
| | | 4 | Acc | 59.71 | 73.35 | 76.04 | 76.35 | 76.38 | - |
| | | | Energy ratio | 0.14 | 0.24 | 0.37 | 0.53 | 0.76 | - |
| | | 8 | Acc | 65.23 | 74.68 | 76.17 | 76.37 | - | - |
| | | | Energy ratio | 0.18 | 0.34 | 0.55 | 0.75 | - | - |
| VGG16 / 138M | 73.25 | 1 | Acc | 0.08 | 0.16 | 1.03 | 55.48 | 72.04 | 73.13 |
| | | | Energy ratio | 0.03 | 0.06 | 0.13 | 0.19 | 0.29 | 0.40 |
| | | 4 | Acc | 70.69 | 72.72 | 73.17 | 73.26 | 73.24 | - |
| | | | Energy ratio | 0.10 | 0.15 | 0.22 | 0.29 | 0.38 | - |
| | | 8 | Acc | 72.26 | 73.16 | 73.22 | 73.22 | - | - |
| | | | Energy ratio | 0.15 | 0.28 | 0.37 | 0.45 | - | - |

## L. Effectiveness of Differential Coding

Table 5 presents a comparative experiment between differential coding and rate coding. In most cases, differential coding outperforms rate coding in both accuracy and energy ratio, particularly as $n$ increases.

## M. Effectiveness of Threshold Iteration Method

Table 6 presents a comparative experiment between the threshold iteration method and the 99.9% large activation method. The threshold iteration method outperforms the 99.9% large activation method across different threshold numbers $n$ and threshold scales.

## N. Evaluation results of object detection task on the COCO dataset

We evaluated the performance of our approach for object detection task on the COCO dataset using three different models provided by torchvision in various parameter settings, along with ablation studies, as shown in Table 7 and 8. The result shows that both differential coding and Threshold Iteration method improves the network's performance.

## O. Evaluation results of Semantic segmentation task on the PascalVOC dataset

Additionally, we evaluated our method for semantic segmentation task on the PascalVOC dataset using two different models provided by torchvision in various parameter settings, also conducting ablation experiments, as presented in Table 9 and 10. The result shows that both differential coding and Threshold Iteration method improves the network's performance.

## P. Algorithm of MT Neuron on GPU

To enable fast execution on the GPU, we also design an efficient algorithm for each time step, as illustrated in Algorithm 3. This algorithm leverages the `torch.float32` data type and takes advantage of the IEEE 754 single-precision floating-point format, where the exponential part has a bias value of 127, as an example.

---

**Algorithm 3** Algorithm of MT Neuron on GPU

---

1: **Input:** Total input $x$ and membrane potential $m$ of MT neuron, number of thresholds parameter $n$.
2: **Output:** Output sum $V_{th}[i] \cdot S[t]$ which is denote as $spike\_sum$
3: **Step 1: Add input to membrane potential**
4: `m = m + x`
5: **Step 2: Set mantissa to zero**
6: `int_tensor = (m*4/3).view(torch.int32)`
7: `mantissa_mask = (1 << 23) - 1`
8: `int_tensor = int_tensor & ~mantissa_mask`
9: **Step 3: Extract the exponential part of spike**
10: `exponent_mask = 0xFF << 23`
11: `spike_exponent = (int_tensor & exponent_mask) >> 23`
12: **Step 4: Find the appropriate threshold to output**
13: `spike_exponent = torch.where(spike_exponent - 127 <= -n, torch.tensor(0,),`
    `torch.where(spike_exponent - 127 > 0, torch.tensor(127,),spike_exponent))`
14: **Step 5: Construct a new exponential part to construct the spike output sum**
15: `int_tensor = (int_tensor & ~exponent_mask) | (spike_exponent << 23)`
16: `spike_sum = int_tensor.view(torch.float32)`
17: **Step 6: Reset membrane potential**
18: `m = m - spike_sum`

---

This algorithm leverages the IEEE 754 float32 format to efficiently compute spike outputs in the neuron model. At each time step, input is added to the membrane potential $m$, which is then scaled and cast to 'int32' to access the exponent field directly.

By masking out the mantissa and extracting the exponent, the algorithm quickly determines the magnitude of $m$. Using the bias value 127 and a threshold parameter $n$, it determines which threshold to apply for generating a spike. The exponent is adjusted accordingly and used to reconstruct 'spike_sum' as a float, which is then subtracted from $m$ to complete the reset.

This approach avoids conditional branches and enables efficient bitwise operations and vectorization on the GPU, making it suitable for high-speed MT neuron simulations.

*Table 4.* Accuracy and Energy Efficiency of DCGS(Ours) on Transformer-based models for ImageNet Dataset

| Architecture/Parameter(M) | Original(ANN)(%) | n | Accuracy Energy | Time-step $T$ | | | | | |
|---|---|---|---|---|---|---|---|---|---|
| | | | | 2 | 4 | 8 | 16 | 32 | 64 |
| ViT-Small / 22.1M | 81.38 | 1 | Acc | 0.1 | 0.1 | 0.1 | 0.11 | 0.19 | 64.92 |
| | | | Energy ratio | 0.00 | 0.001 | 0.008 | 0.06 | 0.28 | 1.34 |
| | | 4 | Acc | 0.1 | 0.16 | 62.76 | 79.25 | 80.95 | - |
| | | | Energy ratio | 0.03 | 0.12 | 0.45 | 1.06 | 2.06 | - |
| | | 6 | Acc | 44.59 | 78.15 | 81.02 | 81.44 | - | - |
| | | | Energy ratio | 0.20 | 0.44 | 0.83 | 1.44 | - | - |
| | | 8 | Acc | 77.84 | 81.11 | 81.43 | 81.38 | - | - |
| | | | Energy ratio | 0.32 | 0.62 | 1.05 | 1.71 | - | - |
| ViT-Base / 86.6M | 84.54 | 4 | Acc | 0.10 | 0.12 | 29.36 | 80.78 | 83.70 | - |
| | | | Energy ratio | 0.01 | 0.05 | 0.28 | 0.54 | 0.74 | 1.44 |
| | | 6 | Acc | 0.10 | 46.03 | 82.72 | 84.69 | - | - |
| | | | Energy ratio | 0.05 | 0.20 | 0.52 | 1.04 | - | - |
| | | 8 | Acc | 80.34 | 83.98 | 84.23 | 84.27 | - | - |
| | | | Energy ratio | 0.28 | 0.54 | 0.92 | 1.46 | - | - |
| ViT-Large / 304.3M | 85.84 | 4 | Acc | 0.10 | 0.10 | 0.18 | 80.74 | 85.00 | - |
| | | | Energy ratio | 0.00 | 0.01 | 0.09 | 0.45 | 0.92 | - |
| | | 6 | Acc | 0.12 | 78.99 | 84.76 | 85.59 | - | - |
| | | | Energy ratio | 0.06 | 0.23 | 0.51 | 0.94 | - | - |
| | | 8 | Acc | 83.73 | 85.45 | 85.68 | 85.74 | - | - |
| | | | Energy ratio | 0.24 | 0.46 | 0.80 | 1.33 | - | - |
| EVA02-Tiny / 5.8M | 80.63 | 4 | Acc | 0.09 | 0.15 | 0.88 | 65.75 | 78.46 | - |
| | | | Energy ratio | 0.01 | 0.05 | 0.24 | 0.94 | 2.30 | - |
| | | 6 | Acc | 0.32 | 52.86 | 77.72 | 80.04 | - | - |
| | | | Energy ratio | 0.11 | 0.34 | 0.86 | 1.81 | - | - |
| | | 8 | Acc | 66.32 | 79.56 | 80.38 | 80.578 | - | - |
| | | | Energy ratio | 0.29 | 0.66 | 1.26 | 2.28 | - | - |
| EVA02-Small / 22.1M | 85.73 | 4 | Acc | 0.09 | 0.10 | 0.14 | 34.86 | 82.66 | - |
| | | | Energy ratio | 0.01 | 0.04 | 0.19 | 0.67 | 1.98 | - |
| | | 6 | Acc | 0.14 | 30.83 | 82.20 | 85.48 | - | - |
| | | | Energy ratio | 0.09 | 0.29 | 0.80 | 1.71 | - | - |
| | | 8 | Acc | 71.37 | 84.70 | 85.64 | 85.72 | - | - |
| | | | Energy ratio | 0.28 | 0.63 | 1.21 | 2.17 | - | - |
| EVA02-Base / 87.1M | 88.69 | 6 | Acc | 3.25 | 81.00 | 87.86 | - | - | - |
| | | | Energy ratio | 0.11 | 0.36 | 0.82 | - | - | - |
| | | 8 | Acc | 84.62 | 88.16 | 88.46 | - | - | - |
| | | | Energy ratio | 0.30 | 0.64 | 1.17 | - | - | - |
| EVA02-Large / 305.1M | 90.05 | 6 | Acc | 12.41 | 87.02 | 89.57 | - | - | - |
| | | | Energy ratio | 0.13 | 0.39 | 0.84 | - | - | - |
| | | 8 | Acc | 88.25 | 89.72 | 89.90 | - | - | - |
| | | | Energy ratio | 0.31 | 0.64 | 1.15 | - | - | - |

*Table 5.* Effective of Differential Coding

| Architecture@ Original(ANN)(%) | Coding Type@ × Threshold scale | n | Accuracy Energy | Time-step $T$ | | | | | |
|---|---|---|---|---|---|---|---|---|---|
| | | | | 2 | 4 | 8 | 16 | 32 | 64 |
| ResNet34@76.42 | Differential@×1 | 1 | Acc | 0.10 | 0.14 | 0.46 | 8.76 | 58.86 | 74.11 |
| | | | Energy ratio | 0.04 | 0.09 | 0.19 | 0.34 | 0.61 | 0.97 |
| | | 4 | Acc | 59.71 | 73.35 | 76.04 | 76.35 | 76.38 | - |
| | | | Energy ratio | 0.14 | 0.24 | 0.37 | 0.53 | 0.76 | - |
| | | 8 | Acc | 65.23 | 74.68 | 76.17 | 76.37 | - | - |
| | | | Energy ratio | 0.18 | 0.34 | 0.55 | 0.75 | - | - |
| | Differential@×2 | 1 | Acc | 0.10 | 0.13 | 0.31 | 2.78 | 46.29 | 73.07 |
| | | | Energy ratio | 0.02 | 0.05 | 0.13 | 0.25 | 0.46 | 0.77 |
| | | 4 | Acc | 46.1 | 69.53 | 75.77 | 76.33 | 76.43 | - |
| | | | Energy ratio | 0.11 | 0.20 | 0.32 | 0.48 | 0.71 | - |
| | | 8 | Acc | 72.03 | 76.24 | 76.39 | 76.41 | - | - |
| | | | Energy ratio | 0.17 | 0.32 | 0.50 | 0.66 | - | - |
| | Rate@×1 | 1 | Acc | 0.11 | 0.29 | 11.03 | 52.78 | 68.05 | 71.04 |
| | | | Energy ratio | 0.04 | 0.11 | 0.26 | 0.55 | 1.11 | 2.22 |
| | | 4 | Acc | 58.46 | 68.46 | 71.20 | 71.77 | 71.99 | - |
| | | | Energy ratio | 0.15 | 0.31 | 0.62 | 1.22 | 2.42 | - |
| | | 8 | Acc | 59.79 | 68.61 | 71.1 | 71.8 | - | - |
| | | | Energy ratio | 0.19 | 0.38 | 0.74 | 1.45 | - | - |
| | Rate@×2 | 1 | Acc | 0.10 | 0.10 | 1.50 | 41.08 | 69.78 | 74.95 |
| | | | Energy ratio | 0.02 | 0.07 | 0.17 | 0.38 | 0.77 | 1.54 |
| | | 4 | Acc | 51.34 | 71.22 | 75.11 | 75.78 | 75.92 | - |
| | | | Energy ratio | 0.13 | 0.26 | 0.53 | 1.05 | 2.09 | - |
| | | 8 | Acc | 65.26 | 74.24 | 75.72 | 75.96 | - | - |
| | | | Energy ratio | 0.19 | 0.46 | 0.73 | 1.43 | - | - |
| ViT-Small@81.38 | Differential@×4 | 1 | Acc | 0.1 | 0.1 | 0.1 | 0.11 | 0.19 | 64.92 |
| | | | Energy ratio | 0.00 | 0.001 | 0.008 | 0.06 | 0.28 | 1.34 |
| | | 4 | Acc | 0.1 | 0.16 | 62.76 | 79.25 | 80.95 | - |
| | | | Energy ratio | 0.03 | 0.12 | 0.45 | 1.06 | 2.06 | - |
| | | 8 | Acc | 77.84 | 81.11 | 81.43 | 81.38 | - | - |
| | | | Energy ratio | 0.32 | 0.62 | 1.05 | 1.71 | - | - |
| | Rate@×4 | 1 | Acc | 0.1 | 0.1 | 0.1 | 0.1 | 0.12 | 54.26 |
| | | | Energy ratio | 0.00 | 0.001 | 0.008 | 0.05 | 0.22 | 1.17 |
| | | 4 | Acc | 0.1 | 0.14 | 51.46 | 78.07 | 80.69 | - |
| | | | Energy ratio | 0.03 | 0.12 | 0.44 | 1.13 | 2.47 | - |
| | | 8 | Acc | 75.64 | 80.29 | 81.18 | 81.36 | - | - |
| | | | Energy ratio | 0.32 | 0.67 | 1.38 | 2.81 | - | - |

*Table 6.* Effective of threshold iteration Method

| Architecture@ Original(ANN)(%) | Find Threshold Method@ ×Threshold Scale | n | Accuracy Energy | Time-step $T$ | | | | | |
|---|---|---|---|---|---|---|---|---|---|
| | | | | 2 | 4 | 8 | 16 | 32 | 64 |
| ResNet34@76.42 | threshold iteration@×1 | 1 | Acc | 0.10 | 0.14 | 0.46 | 8.76 | 58.86 | 74.11 |
| | | | Energy ratio | 0.04 | 0.09 | 0.19 | 0.34 | 0.61 | 0.97 |
| | | 4 | Acc | 59.71 | 73.35 | 76.04 | 76.35 | 76.38 | - |
| | | | Energy ratio | 0.14 | 0.24 | 0.37 | 0.53 | 0.76 | - |
| | | 8 | Acc | 65.23 | 74.68 | 76.17 | 76.37 | - | - |
| | | | Energy ratio | 0.18 | 0.34 | 0.55 | 0.75 | - | - |
| | threshold iteration@×2 | 1 | Acc | 0.10 | 0.13 | 0.31 | 2.78 | 46.29 | 73.07 |
| | | | Energy ratio | 0.02 | 0.05 | 0.13 | 0.25 | 0.46 | 0.77 |
| | | 4 | Acc | 46.1 | 69.53 | 75.78 | 76.33 | 76.43 | - |
| | | | Energy ratio | 0.11 | 0.20 | 0.32 | 0.48 | 0.71 | - |
| | | 8 | Acc | 72.03 | 76.24 | 76.39 | 76.41 | - | - |
| | | | Energy ratio | 0.17 | 0.32 | 0.50 | 0.66 | - | - |
| | 99.9% large activation@×1 | 1 | Acc | 0.10 | 0.14 | 0.74 | 7.08 | 45.68 | 69.50 |
| | | | Energy ratio | 0.04 | 0.09 | 0.19 | 0.35 | 0.64 | 1.08 |
| | | 4 | Acc | 34.08 | 63.18 | 74.41 | 75.82 | 76.14 | - |
| | | | Energy ratio | 0.13 | 0.24 | 0.39 | 0.60 | 0.91 | - |
| | | 8 | Acc | 49.60 | 71.53 | 75.46 | 76.02 | - | - |
| | | | Energy ratio | 0.18 | 0.33 | 0.55 | 0.78 | - | - |
| | 99.9% large activation@×2 | 1 | Acc | 0.10 | 0.10 | 0.22 | 0.93 | 31.32 | 71.36 |
| | | | Energy ratio | 0.02 | 0.05 | 0.13 | 0.25 | 0.50 | 0.89 |
| | | 4 | Acc | 15.74 | 50.97 | 73.88 | 76.12 | 76.32 | - |
| | | | Energy ratio | 0.11 | 0.20 | 0.35 | 0.55 | 0.86 | - |
| | | 8 | Acc | 67.92 | 75.54 | 76.25 | 76.40 | - | - |
| | | | Energy ratio | 0.18 | 0.32 | 0.50 | 0.71 | - | - |

*Table 7.* Accuracy and energy efficiency of DCGS (Ours) across different models for object detection task on the COCO dataset.

| Architecture | ANN mAP%[IoU=0.50:0.95] | mAP Energy ratio | n | Time-step $T$ | | | |
|---|---|---|---|---|---|---|---|
| | | | | 2 | 4 | 6 | 8 |
| FCOS_ResNet50 | 39.2 | mAP | 2 | 0.0 | 0.2 | 1.6 | 6.3 |
| | | Energy ratio | | 0.12 | 0.24 | 0.35 | 0.47 |
| FCOS_ResNet50 | 39.2 | mAP% | 4 | 21.0 | 33.9 | 36.7 | 38.2 |
| | | Energy ratio | | 0.16 | 0.31 | 0.43 | 0.55 |
| FCOS_ResNet50 | 39.2 | mAP% | 8 | 30.5 | 38.5 | 39.2 | 39.2 |
| | | Energy ratio | | 0.22 | 0.42 | 0.61 | 0.75 |
| Retinanet_ResNet50 | 36.4 | mAP% | 8 | 25.6 | 33.9 | 35.8 | 36.0 |
| | | Energy ratio | | 0.23 | 0.44 | 0.63 | 0.78 |
| Retinanet_ResNet50_v2 | 41.5 | mAP% | 8 | 19.7 | 32.6 | 37.9 | 39.7 |
| | | Energy ratio | | 0.22 | 0.43 | 0.64 | 0.84 |

*Table 8.* Ablation Study of DCGS (Ours) on FCOS_ResNet50 model for object detection task on the COCO dataset.

| Coding Type | Threshold Searching method | mAP Energy ratio | n | Time-step $T$ | | | |
|---|---|---|---|---|---|---|---|
| | | | | 2 | 4 | 6 | 8 |
| Differential | Threshold Iteration | mAP% | 8 | 30.5 | 38.5 | 39.2 | 39.2 |
| | | Energy ratio | | 0.22 | 0.42 | 0.61 | 0.75 |
| Rate | Threshold Iteration | mAP% | 8 | 21.8 | 31.5 | 34.3 | 35.5 |
| | | Energy ratio | | 0.22 | 0.44 | 0.66 | 0.88 |
| Differential | 99.9% Large Activation | mAP% | 8 | 25.8 | 36.2 | 38.4 | 39.0 |
| | | Energy ratio | | 0.22 | 0.43 | 0.62 | 0.78 |

*Table 9.* Accuracy and energy efficiency of DCGS (Ours) across different models for semantic segmentation task on the PascalVOC dataset.

| Architecture | ANN mIoU% | mIoU Energy ratio | n | Time-step $T$ | | | |
|---|---|---|---|---|---|---|---|
| | | | | 2 | 4 | 6 | 8 |
| FCN_ResNet50 | 64.2 | mIoU% | 2 | 4.0 | 10.1 | 19.8 | 36.0 |
| | | Energy ratio | | 0.03 | 0.10 | 0.15 | 0.22 |
| FCN_ResNet50 | 64.2 | mIoU% | 4 | 51.8 | 60.5 | 62.7 | 64.0 |
| | | Energy ratio | | 0.10 | 0.20 | 0.27 | 0.35 |
| FCN_ResNet50 | 64.2 | mIoU% | 8 | 61.0 | 64.3 | 64.6 | 64.5 |
| | | Energy ratio | | 0.18 | 0.34 | 0.50 | 0.63 |
| Deeplabv3_ResNet50 | 69.3 | mIoU% | 8 | 66.6 | 69.1 | 69.3 | 69.3 |
| | | Energy ratio | | 0.08 | 0.32 | 0.46 | 0.58 |

*Table 10.* Ablation Study of DCGS (Ours) on FCN_ResNet50 model for semantic segmentation task on the PascalVOC dataset.

| Coding Type | Threshold Searching Method | mAP Energy ratio | n | Time-step $T$ | | | |
|---|---|---|---|---|---|---|---|
| | | | | 2 | 4 | 6 | 8 |
| Differential | Threshold Iteration | mIoU% | 8 | 61.0 | 64.3 | 64.6 | 64.5 |
| | | Energy ratio | | 0.18 | 0.34 | 0.50 | 0.63 |
| Rate | Threshold Iteration | mIoU% | 8 | 58.2 | 62.9 | 63.7 | 63.9 |
| | | Energy ratio | | 0.18 | 0.37 | 0.54 | 0.71 |
| Differential | 99.9% Large Activation | mIoU% | 8 | 61.2 | 64.3 | 64.5 | 64.4 |
| | | Energy ratio | | 0.18 | 0.35 | 0.51 | 0.64 |

