# OpenReview forum: "Differential Coding for Training-Free ANN-to-SNN Conversion"
_ICML.cc/2025/Conference — ICML 2025 poster_

### Official Review · Reviewer_QRfu · 2025-03-10

**Overall Recommendation:** 2

**Summary:**

In this work, the authors proposed differentiable neural coding. Based on the proposed coding, the authors provided differential graded units, differential spiking neurons, and differential coding for linear layer. According to the authors’ experiments, they could achieve state-of-the-art accuracy on image classification tasks.

**Claims And Evidence:**

The proposed methods are theoretically supported well. Experimental evidence should be added to improve the paper.

**Essential References Not Discussed:**

None

**Experimental Designs Or Analyses:**

Ablation studies on proposed methods are required.

From the experimental results in Table 1, it is judged that the CNN model using ReLU does not need minus vth. Why did MT neurons also use these models?

Analysis of overheads such as proposed methods and MT neurons is required. Also, these overheads should be considered in comparing energy consumption.

In addition to image recognition, the experimental results of object detection, segmentation, etc., if added, will be able to highlight the utility of the proposed method.

**Methods And Evaluation Criteria:**

The most important limitation of the SNN model is its applicability to neuromorphic hardware. SNNs must operate on event-based neuromorphic hardware to ensure low-power operation. From this perspective, the method proposed in this study has major concerns. There are great doubts about whether the proposed differentiable neural coding can be applied to neuromorphic hardware. There are also concerns about the feasibility of implementing the proposed differential graded units, differential spiking neurons, and differential linear layers derived from neural coding on neuromorphic hardware. The proposed method is expected to be difficult to implement on neuromorphic hardware.

MT neurons are more complex to implement than LIF-series neurons. In particular, 2n subtract (x-\lambda_p) operations are required to obtain the argmin of Equation 6. This computational overhead not only offsets the advantages of SNNs operating on neuromorphic hardware, but may also make implementation impossible. How can MT neurons be implemented on neuromorphic hardware? - How can it be applied to LIF?

**Other Comments Or Suggestions:**

- It would be better to add synapse part to Figure 1.
- For convenient comparisons, It would be better to present the accuracy of ANN for each experimental result in Table 1. (Even if it is also in the supplementary)

**Other Strengths And Weaknesses:**

Please refer to the above comments.

**Questions For Authors:**

Please refer to the above comments.

**Relation To Broader Scientific Literature:**

It will help advance neuromorphic computing.

**Theoretical Claims:**

The biggest difference between the existing rate coding (Equation 7) and the proposed coding (Equation 14) is described in the manuscript as “differential coding only updates the encoded activation value when an output spike occurs, rather than decay at each time-step in rate coding.” (line 205~). However, this is based on an incorrect fact. Equation 7 is only an equation to explain the ANN-to-SNN conversion of rate coding, and the actual output of the SNN is a binary spike train. Therefore, as the authors mentioned, the firing rate is not calculated at every time point in the spiking neuron, but binary spike output is generated only when a spike is fired. In this regard, there is no difference between the proposed coding and rate coding. Based on the authors’ claim, it seems that the authors did not use binary spike activation. If so, this greatly worsens the advantages of SNN for event-based computing. If binary spike activation is not used, how can it be utilized in neuromorphic hardware? If it is difficult to utilize in neuromorphic hardware, it seems reasonable to consider the proposed model as a DNN with a new activation function.

- Equation 15 - How can the membrane potential (m^l[t]) and the firing rate (r^(l-1)[t]) of the previous layer be the same?

- Equation 16 - It is not reasonable to use non-linear activation (F) to approximate spiking neurons. It is reasonable to simulate only the behavior of spiking neurons that can be supported by neuromorphic hardware without non-linear activation.

---

> ### Author Rebuttal · Authors · 2025-03-31
>
> Thank you for your thoughtful and very detailed feedback. We are delighted that you find our paper theoretically supported well and the results state-of-the-art. We would like to address your concerns and answer your questions in the following:
>
> ### 1. Answer to  "MT neurons are more complex to implement than LIF-series neurons. In particular, 2n subtract ($x-\lambda_p$) operations are required to obtain the argmin of Equation 6." and "Analysis of overheads such as proposed methods and MT neurons is required."
>
> Thank you for raising these concern. Equation 6 in the article is presented for ease of understanding. In hardware implementation, the argmin module is not used. We have developed a hardware-friendly version of the MT neuron model, which can efficiently map the appropriate threshold using the potential's sign bit and exponent bits at an extremely low cost. **The detailed implementation of MT neuron can be seen in our response to Reviewer Yuhw.** We look forward to your review.
>
> ### 2. Answer to "It seems that the authors did not use binary spike activation." and explanation of  "differential coding only updates the encoded activation value when an output spike occurs, rather than decay at each time-step in rate coding."
>
> We use binary spike activation between neurons, as shown by Equation 6 and the red line in Figure 3a.
> Equations 7 and 12 respectively represent the encoding of the information sequence $x^l[1:t]$ in layer $l$ under rate coding and differential coding. Here, $x$ can be the weighted spike according to Equation 4. In rate coding, even though no spike is fired at a given time step and no explicit additional computation is performed, the meaning $r[t]$ encoded by the sequence $x^l[1:t]$ changes due to the increase in $t$, as Equation 7. Conversely, differential coding ensures that when no spike occurs at time-step $t$, the sequence $x^l[1:t]$ retains the same meaning as $x^l[1:t-1]$.
>
> ### 3. From the experimental results in Table 1, it is judged that the CNN model using ReLU does not need minus vth. Why did MT neurons also use these models?
>
> Does the term " minus vth" in your question refer to negative thresholds? Negative thresholds help minimize excessive spikes, which reduce unevenness errors[1] in conversion error. When converting a CNN model with ReLU, Differential Graded Units are not utilized. Instead, the ReLU activation function is replaced with a specific MT Neuron, which uses an additional mask to dynamically disable certain negative thresholds, ensuring the total output remains positive. Detailed implementation will be included in the appendix.
>
> [1] Optimal ANN-SNN Conversion for High-accuracy and Ultra-low-latency Spiking Neural Networks.
>
> ### 4. Explain Equation 15 - How can the membrane potential ($m^l[t]$) and the firing rate ($r^{l-1}[t]$) of the previous layer be the same?
>
> To unify the representation, we consider the linear layer as an independent layer, rather than just weights between layers. When layer $l-1$ is a linear layer, the actual previous layer should be $l-2$. In this case, $x^{l-1}[t]= W^lx^{l-2}=\sum_iW^l\lambda^ls^{l-2}$. Here, $r^{l-1}[t]$ represents the rate obtained by converting differencial coding back into a rate coding after linear layer.
>
> In Equation 15, we define $m^l[t]=r^{l-1}[t]$ and further derive $m^l[t]=m^l[t-1]+\frac{x^{l-1}[t]}{t}$.
>
> ### 5.Explain Equation 16 - It is not reasonable to use non-linear activation (F) to approximate spiking neurons. It is reasonable to simulate only the behavior of spiking neurons that can be supported by neuromorphic hardware without non-linear activation.
>
> We consider the non-linear activation as a part of the neuron's internal dynamics, separate from the communication between neuron layers. This design ensures that spike communication between neuronal layers remains uninterrupted, making our method feasible for hardware implementation. Furthermore, our Equation 16 specifies the expected output, which can also be approximated using multiple IF neurons to achieve hardware implementation[2].
>
> [2] Spatio-temporal approximation: A training-free snn conversion for transformers.
>
> ### 6. Ablation studies on proposed methods are required. And the experimental results of object detection, segmentation, etc., if added, will be able to highlight the utility of the proposed method.
> The ablation studies comparing differential coding and rate coding, as well as threshold iteration and the 99% large activation method, are detailed in Section 5 and Appendices K and M. If there’s anything further you’d like us to add, please let us know.
>
> **In response to Reviewer Kt6R, we have added new experiments and ablation studies on object detection and semantic segmentation tasks.** We look forward to your review.
>
> ### 7. Refine Figure 1 and Table 1.
>
> Thank you for your suggestion. We will revise Figure 1 to make it more biologically explainable and update Table 1 to ensure it is easier for comparisons.

---

### Official Review · Reviewer_Yuhw · 2025-03-13

**Overall Recommendation:** 3

**Summary:**

This paper introduces a novel differential coding scheme for training-free ANN-to-SNN conversion. The authors propose using time-weighted spikes as incremental updates rather than direct rate representations, significantly reducing energy consumption and spike counts. They detail an algorithmic framework integrating multi-threshold spiking neurons, differential coding for various layers (convolutions, fully connected, Transformers), and a threshold iteration method that optimally sets neuron thresholds under a normal distribution assumption.

**Claims And Evidence:**

The authors claim that differential coding reduces the spike rate and preserves high accuracy in converted SNNs. They support this claim with extensive experiments demonstrating both reduced energy consumption and competitive accuracy compared to baseline methods.

**Essential References Not Discussed:**

The paper covers the main works on ANN-to-SNN conversion and multi-threshold neurons. However, Adaptive Calibration [AAAI 2025] is also a training-free and multi-threshold neuron framework. I think it should be included.

Adaptive Calibration: A Unified Conversion Framework of Spiking Neural Network [AAAI 2025]

**Experimental Designs Or Analyses:**

The experimental designs—using multiple CNN and Transformer benchmarks—are comprehensive. The analysis is carefully presented, with comparisons to standard baselines and ablation studies showing how each component (e.g., differential coding vs. threshold tuning) contributes to overall performance.

**Methods And Evaluation Criteria:**

The proposed methods—differential coding and threshold optimization—are well-suited to ANN-to-SNN conversion tasks. The chosen evaluation criteria (accuracy and energy-related metrics) are appropriate and align with typical benchmarks for spiking networks.

**Other Comments Or Suggestions:**

1. It would be beneficial to include a small ablation experiment on threshold selection methods (e.g., comparing the threshold iteration method to a grid search or calibration-based approach) to show how quickly and accurately each method converges.

2. A dedicated subsection on hardware aspects—whether differential coding might exacerbate or mitigate “burst-like” spiking under certain time scales—would strengthen the paper’s real-world applicability.

**Other Strengths And Weaknesses:**

Strengths:
1. The proposed differential coding framework is novel and well-explained.
2. The threshold iteration method is carefully justified with theoretical derivations.
3. Empirical results are thorough, showing gains in both accuracy and energy efficiency.

Weaknesses:
1. While the authors do address threshold tuning, more empirical comparisons (e.g., grid-based calibration [1]) would help quantify speed and accuracy trade-offs.
2. Discussions on hardware implementation are relatively brief—expanding on how differential coding would map to neuromorphic hardware (especially regarding memory overhead and potential short-time “explosive” accumulations) would clarify practical feasibility.

[1] A free lunch from ann: Towards efficient, accurate spiking neural networks calibration.

**Questions For Authors:**

See Suggestions and Weaknesses

**Relation To Broader Scientific Literature:**

The paper builds on established ANN-to-SNN conversion methods but improves them through more efficient coding strategies.

**Theoretical Claims:**

The authors provide proofs and derivations in the supplemental material. The mathematical steps appear sound.

---

> ### Author Rebuttal · Authors · 2025-03-31
>
> Thank you for your positive and thoughtful comments. We are encouraged that you find our method novel and well-explained, and our empirical results thorough. We would like to address your concerns and answer your questions in the following.
>
> ### 1. Hardware implementation of MT Neuron and discussion about memory overhead and the relationship between differential coding and "burst-like" spiking behavior.
>
> Thank you for your suggestion. We will first show how to implement a hardware friendly MT neuron, and then discuss the relationship between differential coding and "burst-like" spiking behavior.
>
> **Hardware implementation of MT Neuron**:
>
> Compared with previous ANN2SNN methods, the MT neuron is required to transmit an extra index $i$ for the threshold. When implementing the MT neuron on GPUs, two implementations can be considered:
> 1. Sent $V_{th}[i] \cdot S[t]$ to the next layer
> 2. Add an external threshold dimension with $2n$ elements to $S[t]$, set $S[t][i]=1$ and $S[t][j]=0$ for all $j \neq i$. At the same time, an external threshold dimension is added to the weight of the next layer, whose elements are the multi-level thresholds.
>
> For simplicity, we use implementation 1 on GPUs, which is not pure binary but equivalent to implementation 2 with binary outputs. The MT neuron is also compatible with asynchronous computing neuromorphic chips because its outputs are still sparse events. Take the speck chip [1] as an example. The LIF neuron in the convolutional layer in speck chip outputs $(c,x,y)$ to the next layer (refer to Fig S4). When using the MT neuron, the only modification is adding a threshold index, i.e., $(c,x,y, i)$. The computations of the next layer should also be changed with a bit-shift operation on weights (because the threshold is the power of 2 and the multiplication is avoided). After the above modifications, the computation is still asynchronous and event-driven.
>
> The implemention to avoid argmin in Equation 6 in hardware can be desctibed in the following two steps.
>
> **Step1:** Set all the base threshold $\theta^l=1$ and get SNN weights by using the weight normalization strategy[2]. So, all thresholds in MT neuron are:
> $$\lambda^l_i=\begin{cases}
> \frac{1}{2^{i-1}},&1<i\leq n,\\\\
> \frac{-1}{2^{i-n-1}},&n<i\leq 2n.
> \end{cases}$$
> **Step2:**
> We define $\frac{4}{3}m^l[t]=(-1)^{S}2^{E}(1+M)$ with $1$ sign bit ($S$), $8$ exponent bits ($E$), and $23$ mantissa bits ($M$).
> Since the median of $\frac{1}{2^{k-1}}$ and $\frac{1}{2^k}$ is $\frac{3}{4}\frac{1}{2^{k-1}}$, we can easily select the correct threshold index $i$ using $E$ and $S$ of $\frac{4}{3}m^l[t]$, without performing $2n$ subtractions to calculate the argmin in Equation 6 :
> $$\text{MTH}_{\theta,n}(m^{l}[t],i)=\begin{cases}1,&\text{if }\begin{cases}
> i<n,\text{ S}=0\text{ and }i=1-\text{E},\\\\
> i\geq n,\text{ S}=1\text{ and }i-n =1-\text{E},\end{cases}\\\\
> 0,&\text{otherwise}.\end{cases}$$
> The detailed implementation will be included in the final version.
>
> For differential neurons, the memory overhead compared to initial neurons, such as IF or MT neurons, only includes an additional membrane potential. This extra potential is used to adjust the input current as described in Theorem 4.4.
>
> **The relationship between differential coding and "burst-like" spiking behavior**:
>
> In this paper, due to the MT neurons, which select an appropriate threshold index to fire spikes, there is no short-time "explosive" accumulation problem. However, this does not prevent us from discussing the effects of using differential coding and burst coding on other neurons that have at least one negative threshold.
>
> Using differential coding can significantly reduce the short-time "explosive" accumulation problem. Since the goal of ANN-to-SNN conversion is to approximate the activation values of an ANN, for neurons that initially suffered from this problem, the differential information diminishes over time. This gradual reduction ultimately eliminates the accumulation issue.
>
> [1] Spike-based Dynamic Computing with Asynchronous Sensing-Computing Neuromorphic Chip
>
> [2] Conversion of Continuous-Valued Deep Networks to Efficient Event-Driven Networks for Image Classification
>
>
> ### 2. Disscussion on comparision with grid search and calibration-based approach.
>
> Thank you for your suggestion. Due to time constraints, we plan to refine the code for the grid search and calibration-based methods in future work to conduct a more detailed comparison of their accuracy. However, **from a speed perspective**, our method calculates the theoretically optimal thresholds for all network modules—whether at the layer-wise, channel-wise, or neuron-wise level—within seconds after computing the mean and variance. We believe this represents a significant improvement in speed compared to grid search and calibration-based methods.
>
>
> ### 3. Incomplete related works.
>
> Thank you for your suggestion. We will cite {Adaptive Calibration [AAAI 2025]} and discuss this article in the final version

---

### Official Review · Reviewer_Kt6R · 2025-03-13

**Overall Recommendation:** 5

**Summary:**

ANN-to-SNN conversion has been known to produce so-called ‘conversion’ errors. Recent studies proposed methods that can reduce conversion errors, and in this study, the authors propose to improve the earlier studies with a novel algorithm named ‘differential coding’. Specifically, they focus on preventing the decay of the early spikes’ influences on spiking neurons’ outputs and show that a new encoding variable in spiking neurons can prevent neurons forgetting early spikes. The analytical analyses on neurons’ behaviors with differential coding makes this study’s objective clear, and the empirical evaluations are compelling. Given the novelty and potential influence of this newly proposed idea, I think this study may be of great interest to our readers.

**Claims And Evidence:**

The paper proposes novel and interesting algorithms, which are well explained. Further, the empirical evaluations clearly support the utility of   the "differential coding" proposed by the authors.

**Essential References Not Discussed:**

N/A

**Experimental Designs Or Analyses:**

Experimental designs and analyses are all sound.

**Methods And Evaluation Criteria:**

The authors tested differentiable coding on only ImageNet. Since ImageNet is a golden standard for image classification, this may not be a critical issue, but it would be great to see evaluations on a few more datasets to strengthen this study’s message.

**Other Comments Or Suggestions:**

In lines 426-427, the authors write "As shown in Table3, detailed results can be found in Appendix". I think the authors may want to rephrase it for better readability.

**Other Strengths And Weaknesses:**

N/A

**Questions For Authors:**

N/A

**Relation To Broader Scientific Literature:**

They described earlier studies sufficiently well.

**Theoretical Claims:**

I did not find any issues with their description.

---

> ### Author Rebuttal · Authors · 2025-03-31
>
> Thank you for your positive and constructive comments. We are delighted that you find our idea novel, interesting and well explained. We would like to address your concerns and answer your questions in the following.
>
> ## 1. It would be great to see evaluations on a few more datasets to strengthen this study’s message.
> Thank you for your suggestion to evaluate our method on more tasks to strengthen the message. We added the test results of our method on object detection and semantic segmentation tasks.
> ### 1.1 Evaluation results of object detection task on the COCO dataset
> We evaluated the performance of our approach for object detection task on the COCO dataset using three different models provided by torchvision in various parameter settings, along with ablation studies, as shown in the table below. The result shows that both differential coding and Threshold Iteration method improves the network's performance.
>
> **Table R1: Accuracy and energy efficiency of DCGS(Ours) across different models for object detection task on the COCO dataset.**
> | Architecture  | ANN mAP%\[IoU=0.50:0.95\]  \/energy ratio | n | T=2   | T=4   | T=6  | T=8 |
> | -------- | -------------- | ----- | ----- | ----- | ----- | ----- |
> | FCOS_ResNet50| mAP%: 39.2| 2 | 0.0 | 0.2 | 1.6 | 6.3 |
> |              | energy ratio| 2 | 0.12 | 0.24 | 0.35 | 0.47 |
> | FCOS_ResNet50 | mAP%: 39.2 | 4 | 21.0 | 33.9 | 36.7 | 38.2 |
> |              | energy ratio    | 4 | 0.16 | 0.31 | 0.43 | 0.55 |
> | FCOS_ResNet50 | mAP%: 39.2 | 8 | 30.5 | 38.5 | 39.2 | 39.2 |
> |              | energy ratio    | 8 | 0.22 | 0.42 | 0.61 | 0.75 |
> | Retinanet_ResNet50| mAP%: 36.4| 8 | 25.6 | 33.9 | 35.8 | 36.0 |
> |              | energy ratio| 8 | 0.23 | 0.44 | 0.63 | 0.78 |
> | Retinanet \_ResNet50_v2| mAP%: 41.5| 8 | 19.7 | 32.6 | 37.9 | 39.7 |
> |              | energy ratio| 8 | 0.22 | 0.43 | 0.64 | 0.84 |
>
> **Table R2: Ablation Study of DCGS(Ours) on FCOS_ResNet50 model for object detection task on the COCO dataset.**
> | Coding Type | Threshold Searching method | mAP% \/energy ratio | n | T=2   | T=4   | T=6  | T=8 |
> | -------- | -------------- | ----- |----- | ----- | ----- | ----- | ----- |
> | Differential| Threshold Iteration |mAP%| 8 | 30.5 | 38.5 | 39.2 | 39.2 |
> |              | |energy ratio| 8 | 0.22 | 0.42 | 0.61 | 0.75 |
> | Rate| Threshold Iteration |mAP%| 8 | 21.8 | 31.5 | 34.3 | 35.5 |
> |              | |energy ratio| 8 | 0.22 | 0.44 | 0.66 | 0.88 |
> | Differential| 99.9% Large Activation |mAP%| 8 | 25.8 | 36.2 | 38.4 |39.0 |
> |              | |energy ratio| 8 | 0.22 | 0.43 | 0.62 | 0.78 |
>
> ### 1.2 Evaluation results of Semantic segmentation task on the PascalVOC dataset
> Additionally, we evaluated our method for semantic segmentation task on the PascalVOC dataset using two different models provided by torchvision in various parameter settings, also conducting ablation experiments, as presented in the table below. The result shows that both differential coding and Threshold Iteration method improves the network's performance.
>
> **Table R3: Accuracy and energy efficiency of DCGS(Ours) across different models for semantic segmentation task on the PascalVOC dataset.**
> | Architecture  | ANN mIoU% \/energy ratio  | n    | T=2   | T=4   | T=6  | T=8 |
> | -------- | -------------- | ----- | ----- | ----- | ----- | ----- |
> | FCN_ResNet50| mIoU%: 64.2 | 2 | 4.0 | 10.1 | 19.8 | 36.0 |
> |    | energy ratio| 2 | 0.03 | 0.10 | 0.15 | 0.22 |
> | FCN_ResNet50 | mIoU%: 64.2 | 4 | 51.8 | 60.5 | 62.7 | 64.0 |
> |    | energy ratio    | 4 | 0.10 | 0.20 | 0.27 | 0.35 |
> | FCN_ResNet50 | mIoU%: 64.2 | 8 | 61.0 | 64.3 | 64.6 | 64.5 |
> |    | energy ratio   | 8 | 0.18 | 0.34 | 0.50 | 0.63 |
> | Deeplabv3_ResNet50| mIoU%: 69.3 | 8  | 66.6 | 69.1 | 69.3 | 69.3 |
> |    | energy ratio| 8 | 0.08 | 0.32 | 0.46 | 0.58 |
>
>
> **Table R4: Ablation Study of DCGS(Ours) on FCN_ResNet50 model for semantic segmentation task on the PascalVOC dataset.**
> | Coding Type | Threshold Searching method | mIoU% \/energy ratio | n | T=2   | T=4   | T=6  | T=8 |
> | -------- | -------------- |----- | ----- | ----- | ----- | ----- | ----- |
> | Differential| Threshold Iteration |mIoU%| 8 | 61.0 | 64.3 | 64.6 | 64.5 |
> |    | |energy ratio| 8 | 0.18 | 0.34 | 0.50 | 0.63 |
> | Rate| Threshold Iteration |mIoU%| 8 | 58.2 | 62.9 | 63.7 | 63.9 |
> |    | |energy ratio| 8 | 0.18 | 0.37 | 0.54 | 0.71 |
> | Differential| 99.9% Large Activation |mIoU%| 8 | 61.2 | 64.3 | 64.5 | 64.4 |
> |    | |energy ratio| 8 | 0.18 | 0.35 | 0.51 | 0.64 |
>
> ## 2. In lines 426-427, the authors write "As shown in Table3, detailed results can be found in Appendix". I think the authors may want to rephrase it for better readability.
> Thank you for your suggestion. We will revise this sentence to "The partial results are presented in Table 3, with a more detailed table provided in Appendix L.". What's more, in the final version, we will reshape Table 3 and Table 5 into line charts to more intuitively compare the different methods.

---

### Decision · Program_Chairs · 2025-05-01

**Decision:**

Accept (poster)

**Comment:**

The authors propose a so-called differential coding algorithm for converting ANN networks to spiking neural networks (SNN), by transmitting changes to the rate rather than the actual spike count.
The authors also present a threshold iteration method for optimizing the spiking thresholds when converting ReLUs to spiking neurons.
The authors do various experiments on CNNs and Transformer architectures and argue that the algorithms
can improve or maintain accuracy while reducing energy consumption.
Overall the paper received 3 reviews, 2 of which were positive and 1 was slightly negative (weak reject).
The positive comments were related to the novelty of the approach, the theoretical justification for the algorithm and the thoroughness of the tests.
One of the reviewers questioned the applicability of the algorithm to current neuromorphic hardware architectures and questioned the efficiency of the proposed neuron model. There was no response to authors rebuttal related to this comment.
Overall I am leaning towards accepting the paper, but I suggest the authors take advantage of the reviewer comments to improve future iterations of the paper.